# Loss of TET2 and TET3 in regulatory T cells unleashes effector function

Xiaojing Yue[1], Chan-Wang J. Lio[1], Daniela Samaniego-Castruita[1], Xiang Li[1] & Anjana Rao[1,2,3]

TET enzymes oxidize 5-methylcytosine to 5-hydroxymethylcytosine and other oxidized methylcytosines in DNA. Here we examine the role of TET proteins in regulatory T (Treg) cells. $Tet2/3^{fl/fl}Foxp3^{Cre}$ mice lacking $Tet2$ and $Tet3$ in Treg cells develop inflammatory disease, and Treg cells from these mice show altered expression of Treg signature genes and upregulation of genes involved in cell cycle, DNA damage and cancer. In littermate mice with severe inflammation, both $CD4^+Foxp3^+$ and $CD4^+Foxp3^-$ cells show strong skewing towards Tfh/Th17 phenotypes. Wild-type Treg cells in mixed bone marrow chimeras and in $Tet2/3^{fl/fl}Foxp3^{WT/Cre}$ heterozygous female mice are unable to rescue the aberrant properties of $Tet2/3^{fl/fl}Foxp3^{Cre}$ Treg cells. Treg cells from $Tet2/3^{fl/fl}Foxp3^{Cre}$ mice tend to lose Foxp3 expression, and transfer of total $CD4^+$ T cells isolated from $Tet2/3^{fl/fl}Foxp3^{Cre}$ mice could elicit inflammatory disease in fully immunocompetent mice. Together, these data indicate that $Tet2$ and $Tet3$ are guardians of Treg cell stability and immune homeostasis.

[1] Division of Signaling and Gene Expression, La Jolla Institute for Immunology, La Jolla, CA 92037, USA. [2] Sanford Consortium for Regenerative Medicine, La Jolla, CA 92037, USA. [3] Department of Pharmacology and Moores Cancer Center, University of California, San Diego, La Jolla, CA 92093, USA. Correspondence and requests for materials should be addressed to A.R. (email: arao@lji.org)

Regulatory T (Treg) cells are a distinct lineage of CD4[+] T lymphocytes that prevent autoimmunity against self-antigens and maintain immune homeostasis[1,2]. The lineage specification, development, and suppressive function of Treg cells are controlled by Foxp3, an X-chromosome-encoded transcription factor that is the most specific marker distinguishing Treg cells from other T helper cell populations[3]. Foxp3-expressing Treg cells have been reported to be stable in both steady state and inflammatory conditions in vivo[4], but a fraction of Treg cells show unstable Foxp3 expression[5,6]. Fate mapping experiments have shown that many of these Treg cells are "ex-Treg" cells that derive from Foxp3-expressing Treg cells that subsequently lose Foxp3 expression[5], but some may be conventional T cells that transiently express Foxp3, thus representing a minor population of uncommitted Treg cells[6]. Moreover, Treg-specific genetic ablation of many gene products—including cell surface proteins such as neuropilin1, transcription factors such as NR4A, signaling molecules such as PTEN, and epigenetic regulators such as EZH2—results in instability of Treg Foxp3 expression and/or loss of suppressive function[7–11].

Epigenetic changes, including histone modifications, DNA methylation and chromatin accessibility, play a critical role in establishing and maintaining the Treg cell lineage[3]. The best studied of these epigenetic changes is DNA demethylation at various Treg-specific demethylated regions (TSDRs), including two "conserved non-coding sequences" (CNS) CNS1 and CNS2 within the Foxp3 locus[12,13]. The stability of Foxp3 expression is closely linked to the demethylated status of CNS1 and CNS2[12,13], while Treg-specific gene expression and optimal Treg function are linked to the DNA methylation status of other relevant TSDRs. Thus, it is important to understand how DNA methylation is regulated at these elements, and how it in turn controls the stability and function of Treg cells.

TET family proteins are Fe(II) and 2-oxoglutarate-dependent dioxygenases that catalyze oxidation of the methyl group of 5-methylcytosine (5mC) in DNA to 5-hydroxymethylcytosine (5hmC) and the further oxidized products 5-formylcytosine (5fC) and 5-carboxylcytosine (5caC)[14–18]. These modified bases are intermediates in DNA demethylation[19–21]. TET proteins and DNA modification exert a pervasive influence on immune/hematopoietic cell development and function. Acute deletion of Tet2 and Tet3 in hematopoietic stem cells induced the rapid development of an aggressive and fully-penetrant myeloid leukemia in adult mice[22]. Deletion of Tet2 and Tet3 by Mb1-Cre in early B cells resulted in developmental blockade at the pro-B to pre-B cell transition due to a defect in immunoglobulin light chain rearrangement[23,24]. Deletion of Tet2 and Tet3 in T cells mediated by CD4-Cre led to an antigen-driven expansion of invariant NKT (iNKT) cells, which developed rapidly into CD1d-restricted iNKT cell lymphoma[25]. Treg cells in this Tet2/3-deficient mouse strain displayed unstable Foxp3 expression, concomitantly with DNA hypermethylation at CNS1, CNS2, and other TSDRs[12]. Combined deletion of Tet1 and Tet2 also resulted in CNS2 hypermethylation and impaired Treg cell differentiation and function[26].

Our previous study on the role of TET proteins in Treg cells[12] was complicated by the iNKT cell expansion occurring in the same mouse strain, in which Tet gene deletion was mediated by CD4-Cre[12,25]. Here, we used a mouse strain in which Tet2 and Tet3 deficiency were targeted specifically to Foxp3-expressing Treg cells using Foxp3$^{YFP-Cre}$ (Foxp3$^{Cre}$). Tet2/3$^{fl/fl}$Foxp3$^{Cre}$ mice develop an inflammatory disease with splenomegaly and leukocyte infiltration into lung, and CD4[+]Foxp3[+] Treg cells, CD4[+]Foxp3[−] and CD8[+] T cells in these mice display an activated phenotype. Tet2/3$^{fl/fl}$Foxp3$^{Cre}$ Treg cells show dysregulation of Treg signature genes and genes related to cell cycle, DNA damage

and cancer compared to WT Treg cells. Perplexingly, a very similar inflammatory disease develops in Tet2/3$^{fl/fl}$Foxp3$^{WT/Cre}$ heterozygous female mice and in mixed bone marrow chimeras in which lethally irradiated mice were reconstituted with a 1:1 mixture of wild-type and Tet2/3$^{fl/fl}$Foxp3$^{Cre}$ bone marrow cells, indicating that wild-type Treg cells was not sufficient to rescue the inflammatory phenotype observed in Tet2/3 DKO mice. Fate-mapping experiments showed that Treg cells from Tet2/3$^{fl/fl}$Foxp3$^{Cre}$ DKO mice are more prone to lose Foxp3 expression and become "ex-Treg" cells. Furthermore, transfer of total CD4[+] T cells from Tet2/3$^{fl/fl}$Foxp3$^{Cre}$ DKO mice, which contained these ex-Treg cells, elicits inflammatory disease in immunocompetent mice. Thus, TET deficiency in Treg cells resulted in a dominant inflammatory disease, in which the inflammatory phenotype was driven, at least in part, by ex-Treg cells that acquired effector function. Our data emphasize that TET proteins are essential for maintenance of Treg cell stability and immune homeostasis in mice.

## Results

**Tet2/3-deficiency in Treg cells leads to T cell activation.** To investigate the role of TET proteins in Treg cells, we crossed mice carrying LoxP-flanked Tet2 and Tet3 alleles (Tet2/3$^{fl/fl}$) with Foxp3$^{YFP-Cre}$ (Foxp3$^{Cre}$) mice in which YFP-Cre fusion protein is knocked into the 3' UTR of the Foxp3 gene[27], to generate mice with Treg-specific deletion of Tet2 and Tet3 (Tet2/3$^{fl/fl}$Foxp3$^{Cre}$ mice). Tet2 and Tet3 mRNAs were specifically deleted in CD4[+]YFP[+] Treg cells but not in CD4[+]YFP[-] conventional T cells (Supplementary Fig. 1a). Mice lacking Tet2 and Tet3 in Treg cells did not survive past 8–22 weeks of age (Fig. 1a), although a fraction of male mice survived slightly longer than female mice (Supplementary Fig. 1b). Tet2/3$^{fl/fl}$Foxp3$^{Cre}$ mice displayed splenomegaly and lymphadenopathy, primarily of mesenteric lymph nodes (mLNs, Supplementary Fig. 1c), as evidenced by an increased cellularity (Fig. 1b). The slight increase in cellularity observed in peripheral lymph nodes (pLNs) did not reach statistical significance (Fig. 1b). Histological analysis revealed disrupted splenic architecture in Tet2/3$^{fl/fl}$Foxp3$^{Cre}$ mice with expansion of the white pulp areas, accompanied by leukocyte infiltration into the lung (Supplementary Fig. 1d). Examination of peripheral blood showed an increase in neutrophils and a decrease in lymphocytes, which were within the normal range; and the concentration of red blood cells appeared normal (Supplementary Table 1). Tet2/3$^{fl/fl}$Foxp3$^{Cre}$ mice had significantly higher titers of anti-dsDNA antibodies in the serum compared to WT mice (Supplementary Fig. 2a), suggesting altered self-tolerance. In addition, the titer of serum IgG2b isotype was significantly higher in Tet2/3$^{fl/fl}$Foxp3$^{Cre}$ mice than in WT mice, and there was a tendency towards increased titers of serum IgG1, IgG2a, IgG3, and IgM, which correlated with the severity of disease development in the mice. In contrast, the titer of serum IgA appeared slightly decreased in Tet2/3$^{fl/fl}$Foxp3$^{Cre}$ mice compared to WT mice (Supplementary Fig. 2b).

Tet2/3$^{fl/fl}$Foxp3$^{Cre}$ mice displayed a significant reduction in the percentages of CD4[+] and CD8[+] T cells in spleen and pLNs (Fig. 1c, left panel, Supplementary Fig. 3a), but the numbers of CD4[+] and CD8[+] T cells were roughly maintained except for an increase in CD4[+] T cell numbers in the spleen (Fig. 1c, right panel). The percentage of CD62L$^{hi}$CD44$^{lo}$ naïve cells was dramatically decreased in both CD4[+] and CD8[+] T cell populations, indicating that the majority of the T cells were activated or became memory cells (Fig. 1d, e and Supplementary Fig. 3b). We also observed a reduction in the percentage of B cells and an increase in the percentage of CD11b[+]Gr1[+] myeloid cells (Supplementary Fig. 4).

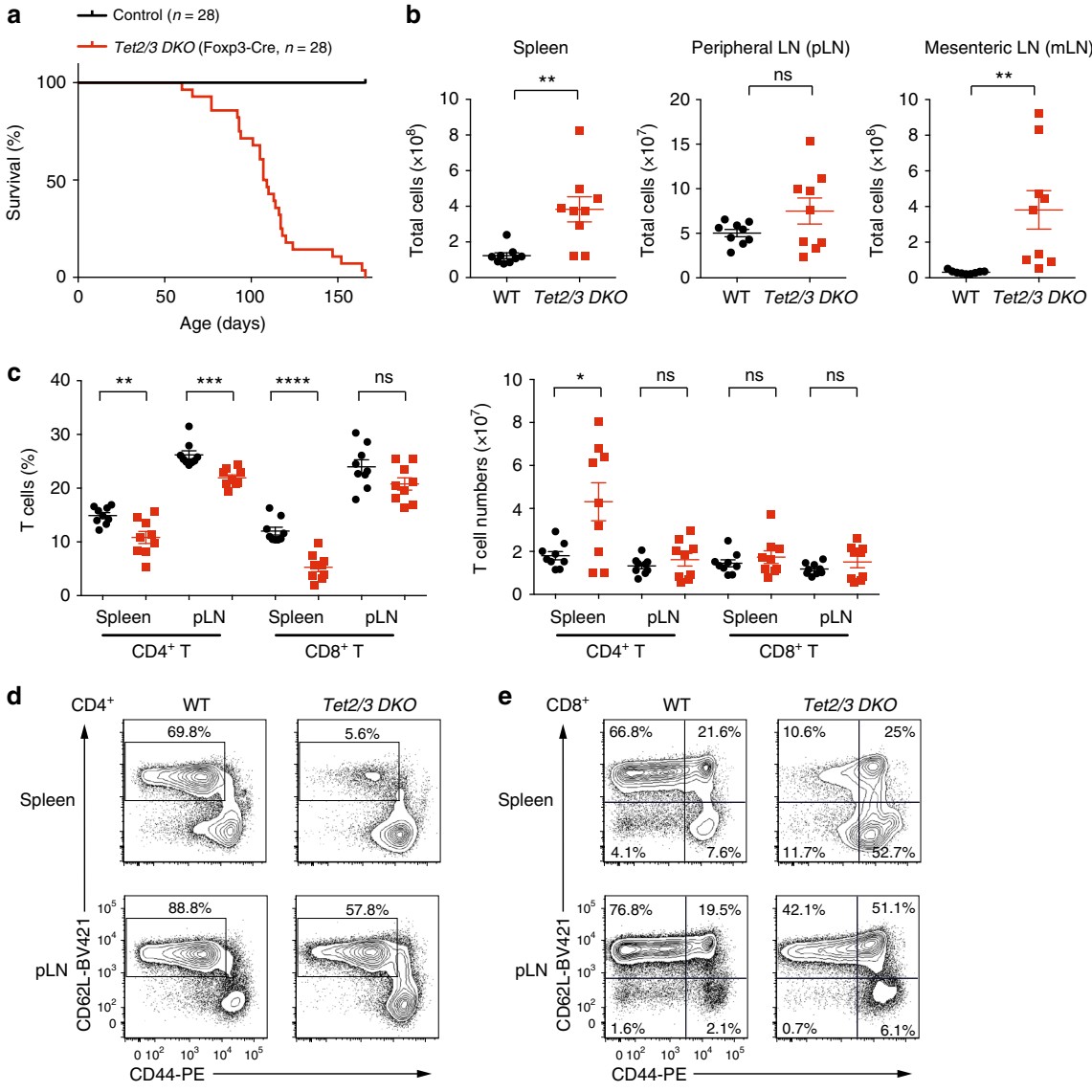

**Fig. 1** Phenotypic analysis of mice with Treg-specific deletion of *Tet2* and *Tet3*. **a** Survival curves for control WT and *Tet2/3*[fl/fl]*Foxp3*[Cre] DKO mice (n = 28). **b** Total cell numbers in spleen, peripheral lymph nodes (pLN) and mesenteric lymph nodes (mLN) of WT and *Tet2/3*[fl/fl]*Foxp3*[Cre] DKO mice (13-16 weeks old, n = 9). **c** Percentages (*left panel*) and cell numbers (*right panel*) of CD4+ and CD8+ T cells in the spleen and pLN from WT and *Tet2/3*[fl/fl]*Foxp3*[Cre] DKO mice (13–16 weeks-old, n = 9). **d** Representative flow cytometry analysis of CD62L and CD44 expression in CD4+Foxp3- T cells from spleen (*upper panels*) and pLN (*lower panels*). **e** Representative flow cytometry analysis of CD62L and CD44 expression in CD8+ T cells from spleen (*upper panels*) and pLN (*lower panels*). Statistical analysis was performed using two-tailed unpaired student's *t* test (*P < 0.05, **P < 0.01, ***P < 0.001, ****P < 0.0001). Error bars show mean ± s.d. from at least three independent experiments

**Dysregulated cell surface phenotype of *Tet2/3 DKO* Treg cells.** We next examined the phenotypic features of *Tet2/3*-deficient Treg cells. The frequency of Foxp3+ Treg cells (as a percentage of total CD4+ T cells) was increased in spleen, as well as in peripheral and mesenteric lymph nodes of *Tet2/3*[fl/fl]*Foxp3*[Cre] mice relative to that of WT mice (Fig. 2a, b). Despite the increase in frequency, *Tet2/3 DKO* Treg cells had lower expression levels of CD25 (IL-2 receptor α chain), a signature surface cytokine receptor of Treg and activated T cells; they also displayed a significant decrease in the expression of neuropilin-1 (Nrp1), an important mediator of Treg homeostasis and function[28], in both spleen and pLNs (Fig. 2c). Flow cytometric analysis revealed varying degrees of increased expression and/or an increased proportion of cells expressing the transcription factor Helios (Ikzf2) and the surface proteins ICOS, CD103, GITR, CTLA-4, and PD-1 in *Tet2/3 DKO* compared to those in WT Treg cells,

especially in the spleen; but similar expression of CD127, IL-7 receptor α chain (Fig. 2d). Both Nrp1 and Helios have been reported to be markers of thymic-derived Treg cells and were used to distinguish thymic-derived Treg cells from peripherally-induced Treg cells;[29–31] however, the reliability of these markers is still controversial[32,33]. The expression of Nrp1 was downregulated in Treg cells from mLNs in *Tet2/3*[fl/fl]*Foxp3*[Cre] mice compared to that of WT mice, while the expression of Helios was increased (Supplementary Fig. 3c). The chemokine receptor CCR7, which is important for the migration and function of Treg cells[34], was also significantly downregulated in Treg cells from *Tet2/3*[fl/fl]*Foxp3*[Cre] compared to WT mice (Supplementary Fig. 3d). Moreover, compared to their WT counterparts, Treg cells from *Tet2/3*[fl/fl]*Foxp3*[Cre] mice showed a decreased proportion of CD62L[hi]CD44[lo] cells (Fig. 2e) and a higher frequency of CD69+ recently activated cells (Fig. 2f), indicating that *Tet2* and

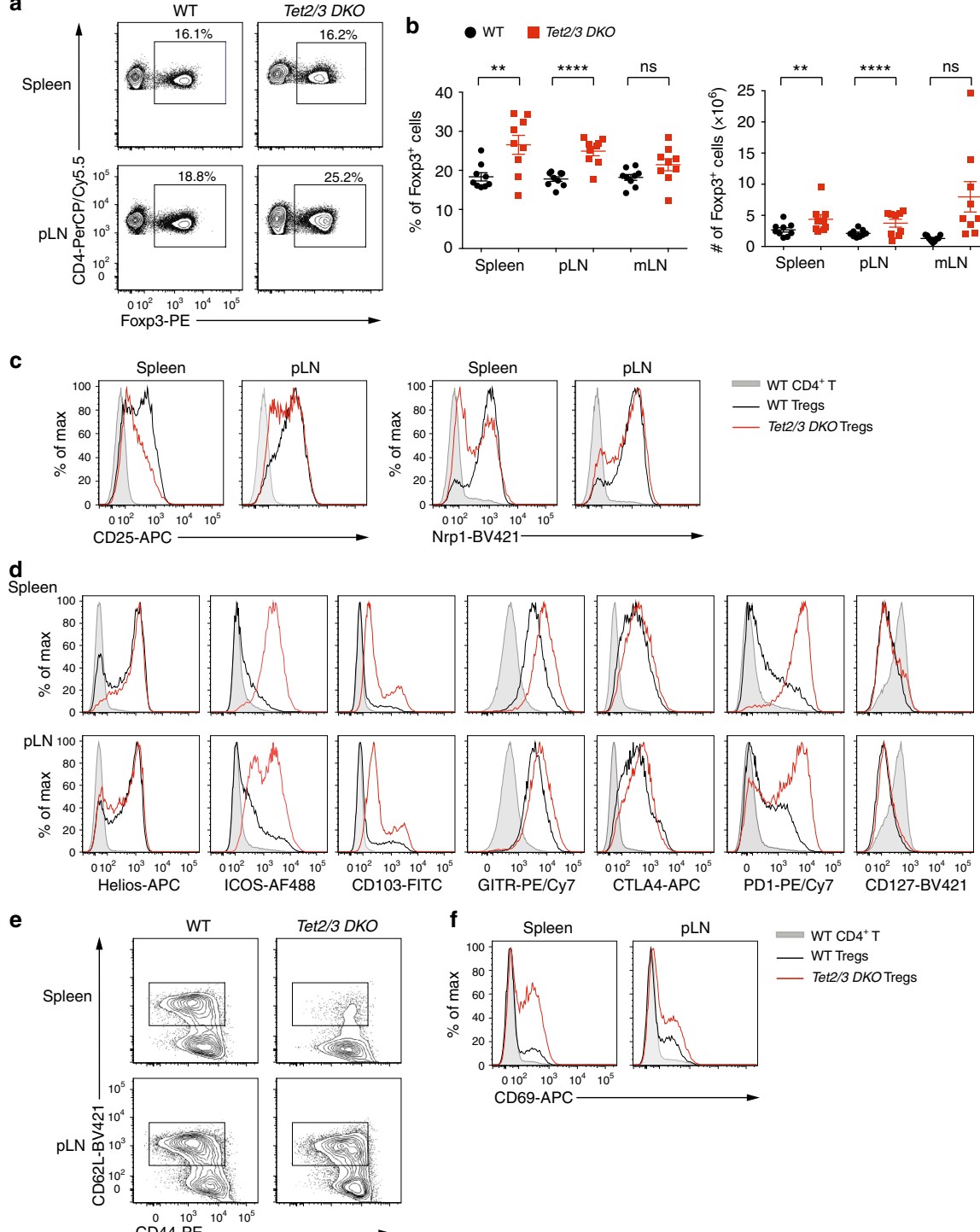

**Fig. 2** Characterization of *Tet2/3*-deficient Treg cells. **a** Representative flow cytometry analyses of the percentage of Treg cells (% of Foxp3+ cells in gated CD4+ T cells) in spleen and pLN from WT and *Tet2/3*fl/flFoxp3Cre DKO mice (13–16 weeks-old). **b** Quantification of the percentages (*left panel*) and cell numbers (*right panel*) of Treg cells in spleen, pLN and mLN from WT and *Tet2/3*fl/flFoxp3Cre DKO mice (13-16 weeks old, n = 9). **c–f** Flow cytometry analysis of *Tet2/3*-deficient Treg cells from spleen and pLN for the expression of CD25 and Nrp1 (**c**); Helios, ICOS, CD103, GITR, CTLA4, PD1 and CD127 (**d**); CD62L versus CD44 (**e**); and CD69 (**f**). Shown are WT CD4+ T cells (shaded gray); WT Tregs (black line); DKO Tregs (red line). Statistical analysis was performed using two-tailed unpaired student's *t* test (*P < 0.05, **P < 0.01, ***P < 0.001, ****P < 0.0001). Error bars show mean ± s.d. from at least three independent experiments

*Tet3* deficiency in Treg cells results in an activated phenotype and dysregulated expression of multiple Treg activation and phenotypic molecules. The increased frequency and activation status of Treg cells that we observe in *Tet2/3*fl/flFoxp3Cre mice is also

commonly observed in mice with Treg perturbations, and may reflect a compensatory increase in Treg cell numbers observed under conditions of insufficient overall Treg cell function[8,35].

**Decreased long term suppression by *Tet2/3 DKO* Treg cells**. To investigate the in vivo function of Treg cells isolated from *Tet2/3<sup>fl/fl</sup>Foxp3<sup>Cre</sup>* mice, we used a mouse model in which we attempted to correct the *scurfy* phenotype. *Scurfy* mice lack Treg cells due to a natural frame-shift mutation in the *Foxp3* gene that abrogates Foxp3 protein expression[36]. Transfer of peripheral CD4<sup>+</sup> T cells from *scurfy* male mice into *Rag1*-deficient recipients leads to an expansion of autoreactive *scurfy* CD4<sup>+</sup> T cells and severe autoimmune inflammation due to a lack of effective Treg cell suppression; both features can be suppressed by co-transfer of congenitally marked WT Treg cells[37]. We transferred $5 \times 10^5$ *scurfy* CD4<sup>+</sup> T cells (CD45.1<sup>+</sup>) into *Rag1*-deficient recipient mice together with $1 \times 10^5$ Treg cells from CD45.2<sup>+</sup> mice, either WT *Foxp3<sup>Cre</sup>* or *Tet2/3<sup>fl/fl</sup>Foxp3<sup>Cre</sup>* mice (Supplementary Fig. 5a). In this short-term in vivo suppression assay, Treg cells isolated from *Tet2/3<sup>fl/fl</sup>Foxp3<sup>Cre</sup>* mice suppressed the expansion of *scurfy* CD4<sup>+</sup> T cells to almost the same extent as Treg cells isolated from WT mice, as evidenced by decreased expansion of CD45.1<sup>+</sup> *scurfy* cells measured in spleen and lymph nodes 4–5 weeks after adoptive transfer (percentages, Supplementary Fig. 5b; total cell numbers, Supplementary Fig. 5c). As we previously observed for Treg cells from *Tet2/3<sup>fl/fl</sup>CD4<sup>Cre</sup>* mice[12], Treg cells isolated from *Tet2/3<sup>fl/fl</sup>Foxp3<sup>Cre</sup>* mice were more likely to lose Foxp3 expression compared to WT control Treg cells (Supplementary Fig. 5b, c).

To examine Treg function over a longer time course, we transferred bone marrow cells from *scurfy* mice into sublethally irradiated *Rag1<sup>-/-</sup>* recipient mice, alone or together with bone marrow cells from WT or *Tet2/3<sup>fl/fl</sup>Foxp3<sup>Cre</sup>* mice (Supplementary Fig. 5d, e). As expected[38], recipient mice receiving *scurfy* bone marrow cells alone started to lose weight at 4 weeks and succumbed 6–7 weeks after transfer. Co-transfer of WT bone marrow cells suppressed *scurfy* cell expansion and protected the recipient mice from death, whereas co-transfer of *Tet2/3 DKO* bone marrow cells failed to protect the recipient mice, which lost weight after 9 weeks and died at 11–13 weeks after transfer (Supplementary Fig. 5d, e). Together, these data indicate that *Tet2/3*-deficient Treg cells from *Tet2/3<sup>fl/fl</sup>Foxp3<sup>Cre</sup>* mice have normal in vivo suppressive function in the short-term that is not sustained in the long-term.

**Moderate hypermethylation of *CNS2* in *Tet2/3 DKO* Treg cells**. The expression of *Foxp3* during Treg cell differentiation is regulated by three conserved noncoding sequence (CNS) elements located in the first intron of the *Foxp3* gene[39]. Among these, *CNS1* and *CNS2* have been shown to control the stability of Foxp3 expression in a manner linked to the DNA modification status of these elements[12,13]. Given that TET proteins catalyze the process of loss of 5mC at *CNS1* and *CNS2* during Treg development in vivo[12], we examined the DNA modification status at *Foxp3 CNS1* and *CNS2* in CD4<sup>+</sup>CD25<sup>+</sup>YFP<sup>+</sup> peripheral Treg cells isolated from male WT or *Tet2/3<sup>fl/fl</sup>Foxp3<sup>Cre</sup>* mice using bisulfite-sequencing (BS-seq) (Supplementary Fig. 6a). The results clearly show impaired DNA demethylation (assessed as loss of 5mC) in *Foxp3 CNS1* and *CNS2* in Treg cells isolated from *Tet2/3<sup>fl/fl</sup>Foxp3<sup>Cre</sup>* mice; however, this impairment was less pronounce (i.e., 5mC + 5hmC levels assessed by BS-seq were lower) than we had previously observed in Treg cells isolated from *Tet2/3<sup>fl/fl</sup>CD4<sup>Cre</sup>* mice (Fig. 3a, b). This result is likely to reflect the chronological order of CD4<sup>Cre</sup> versus Foxp3<sup>Cre</sup> expression: CD4<sup>Cre</sup> mediates *Tet2* and *Tet3* deletion at an earlier developmental stage compared to Foxp3<sup>Cre</sup>. Consistent with this difference, CpG sites #6 and #9-11 in the *CNS2* region, which have already initiated the process of loss of 5mC at the Foxp3<sup>+</sup>CD25<sup>-</sup> precursor stage[12], were efficiently demethylated in *Tet2/3<sup>fl/fl</sup>Foxp3<sup>Cre</sup>* Treg cells. We also examined several other

regulatory regions that show Treg-specific hypomethylation: *Il2ra* intron1a, *Tnfrsf18* exon5, *Ikzf4* intron1b, and *Ctla4* exon2[40]. The DNA modification status at *Il2ra* intron1a, *Tnfrsf18* exon5 and *Ikzf4* intron1b showed similar pattern compare to that of *Foxp3 CNS1* and *CNS2* regions: DNA demethylation was impaired in Treg cells isolated from *Tet2/3<sup>fl/fl</sup>Foxp3<sup>Cre</sup>* mice, but to a lesser extent, compare to that in Treg cells isolated from *Tet2/3<sup>fl/fl</sup>CD4<sup>Cre</sup>* mice (Fig. 3c–e). However, at *Ctla4* exon2, the levels of impairment in DNA demethylation were similar regardless of whether Treg cells were isolated from *Tet2/3<sup>fl/fl</sup>Foxp3<sup>Cre</sup>* mice or *Tet2/3<sup>fl/fl</sup>CD4<sup>Cre</sup>* mice (Fig. 3f).

**Altered gene expression patterns in *Tet2/3 DKO* Treg cells**. To further investigate the molecular mechanisms by which TET proteins control Treg cell identity and function, we compared the transcriptional profiles of CD4<sup>+</sup>YFP<sup>+</sup> Treg cells sorted from 14-weeks-old *Foxp3<sup>Cre</sup>* WT and *Tet2/3<sup>fl/fl</sup>Foxp3<sup>Cre</sup>* DKO mice (2 biological replicates each, isolated from pooled spleen and pLNs) (Supplementary Fig. 6b). We isolated RNA from two *Tet2/3<sup>fl/fl</sup>Foxp3<sup>Cre</sup>* DKO littermates with the assumption that these mice would have comparable phenotypes. However, when sacrificed, one of these mice (labeled with an asterisk: DKO*) clearly showed more severe splenomegaly than its littermate (DKO). This difference was obvious in principal component analysis (PCA) of the RNA-seq data: the biological replicates of WT cells, whether Treg or CD4<sup>+</sup>Foxp3<sup>-</sup> cells, clustered closely together, while those from Treg and CD4<sup>+</sup>Foxp3<sup>-</sup> cells obtained from the two 14-weeks-old *Tet2/3<sup>fl/fl</sup>Foxp3<sup>Cre</sup>* littermates were not as closely aligned (Supplementary Fig. 7a). As expected, both Treg and CD4<sup>+</sup>Foxp3<sup>-</sup> T cells from the *Tet2/3<sup>fl/fl</sup>Foxp3<sup>Cre</sup>* mouse with less severe splenomegaly more closely resembled the corresponding WT cells, whereas cells from the mouse with more severe splenomegaly were less similar to WT (Supplementary Fig. 7a). Nevertheless, scatter plots of the data showed reasonably good correlation between the two replicates from *Tet2/3 DKO* Treg and CD4<sup>+</sup>Foxp3<sup>-</sup> T cells (0.92 and 0.90, respectively), although the correlation between the WT replicates was stronger (0.94 and 0.93, respectively) (Supplementary Fig. 7b, c). Based on these data, we averaged the two DKO replicates for subsequent analyses.

Based on a cut-off of FDR (false discovery rate) ≤ 0.05 and a fold change ≥ 1.5, 1565 genes were differentially expressed, with 1156 genes being upregulated and 409 genes being downregulated in *Tet2/3 DKO* Treg cells compared to WT Treg cells (Fig. 4a). Seventy-six Treg signature genes were differentially expressed (Supplementary Data 1) and a selected subset (*Foxp3*, *Nrp1*, *Il1rl1*, *Prg4*, *Dnahc7b*, and *Klrg1*) are pointed out in the mean average (MA) plot (Fig. 4a). Ingenuity pathway analysis (IPA) of the differentially expressed genes showed that *Tet2/3* deficiency in Treg cells affected three main categories of canonical pathways (Fig. 4b and Supplementary Data 2): genes involved in DNA damage, DNA repair (*Brca1*, *Atm*, *Rad50*), and cell cycle (*Cdkn1a*, *Cdkn2a*, *Hipk2*) (*green bars*); genes implicated in immune cell function such as the antigen presentation pathway (*H2-Aa*, *H2-Ab*, and *H2-Eb1*), Th1 and Th2 activation pathways, etc (*blue bars*); and genes related to molecular mechanisms of cancer (*orange bars*).

A recent study demonstrated that microbial-mediated inflammatory signals were critical for myeloid expansion of *Tet2*-deficient mice[41]. Therefore, we used RNA-seq to examine the transcriptional profiles of CD4<sup>+</sup>YFP<sup>+</sup> Treg cells sorted from mLNs of 14-week-old WT and *Tet2/3<sup>fl/fl</sup>Foxp3<sup>Cre</sup>* mice (3 biological replicates each) (Supplementary Fig. 6b). There was good correlation among the biological replicates (Supplementary Table 2). Based on a cut-off of FDR (false discovery rate) ≤ 0.05

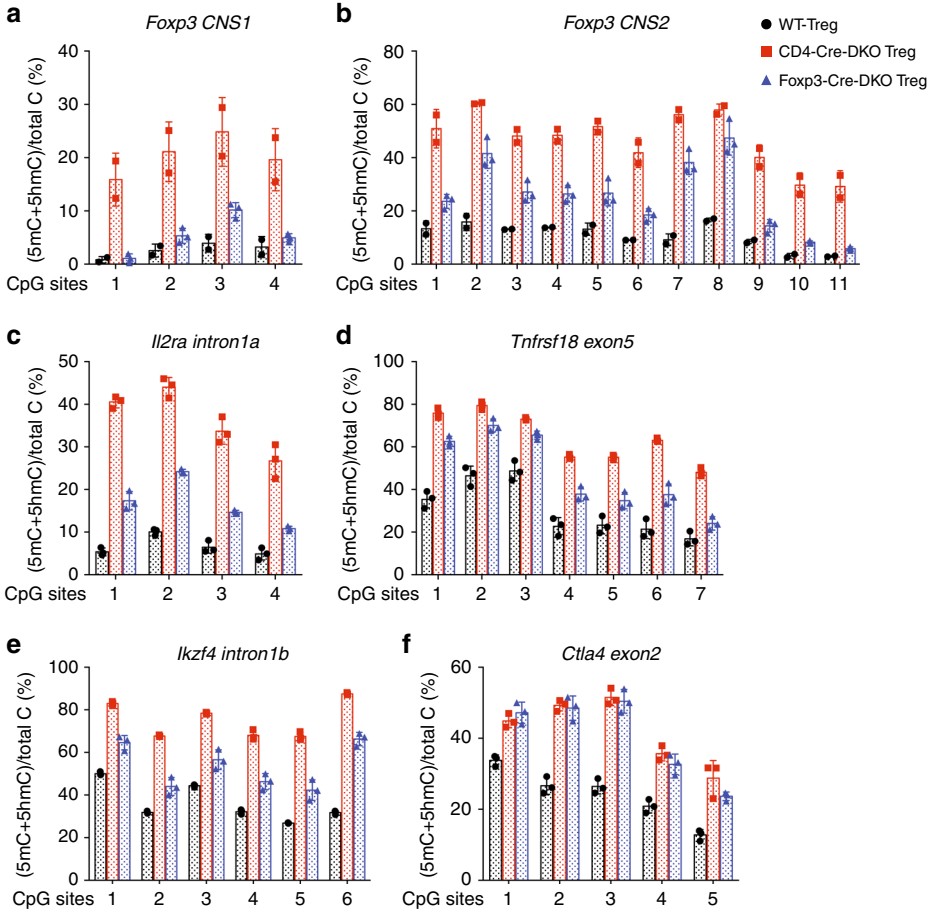

**Fig. 3** DNA methylation status of TSDRs in WT and $Tet2/3^{fl/fl}Foxp3^{Cre}$ DKO Treg cells. **a–f** DNA methylation status of TSDRs (Treg-specific demethylated regions). Graphs depict the percentage of 5mC + 5hmC determined by BS-seq in peripheral Tregs in: (**a**) 4 CpGs in *Foxp3 CNS1*, and (**b**) 11 CpGs in *Foxp3 CNS2*, (**c**) 4 CpGs in *Il2ra intron 1a*, (**d**) 7 CpGs in *Tnfrsf18 exon 5*, (**e**) 6 CpGs in *Ikzf4 intron 1b* and (**f**) 5 CpGs in *Ctla4 exon 2*. Error bars show mean ± s.d. of thousands of sequencing reads from two independent experiments. Wild type (WT) Treg cells (8–10 weeks-old) are shown in black, CD4[Cre] mediated $Tet2/3$ DKO Treg cells (3–4 weeks-old) are shown in red[12] and $Foxp3^{Cre}$ mediated $Tet2/3$ DKO Treg cells (8–10 weeks-old) are shown in blue

and a fold change ≥ 1.5, 2073 genes were differentially expressed, with 1449 genes being upregulated and 624 genes being downregulated in *Tet2/3 DKO* Treg cells compared to WT Treg cells (Supplementary Fig. 8a). A handful of them overlapped with the differentially expressed genes from *Tet2/3 DKO* vs WT Treg cells isolated from pooled spleen and pLNs (Supplementary Fig. 8b). IPA analysis of the differentially expressed genes indicated that similar canonical pathways were affected, including genes involved in cell cycle, DNA damage, immune cell function, and cancer development (Supplementary Fig. 8c and Supplementary Data 3).

The majority of cell cycle-related genes were upregulated in *Tet2/3 DKO* Treg cells (see heatmap of Fig. 4c and Supplementary Fig. 8d), prompting us to assess the proliferation of *Tet2/3 DKO* Treg cells in vivo. We injected WT and $Tet2/3^{fl/fl}Foxp3^{Cre}$ mice with BrdU, a thymidine analog that is incorporated into newly synthesized DNA during replication, and analyzed Treg cells 24 h later. Indeed, *Tet2/3 DKO* Treg cells incorporated significantly more BrdU compared to WT Treg cells (Fig. 4d). The genes involved in DNA damage pathways also prompted us to examine the level of phosphorylated histone H2A.X (γH2AX), an early marker of DNA damage[42], by flow cytometry. In fact, the MFI (mean fluorescent intensity) of γH2AX staining was significantly increased in both Treg cells and CD4+Foxp3− cells in spleen and mLN isolated from $Tet2/3^{fl/fl}Foxp3^{Cre}$ compared to WT mice (Supplementary Fig. 9). But we did not observe any increase in

the level of active caspase-3, an early stage apoptosis marker, in Treg cells from $Tet2/3^{fl/fl}Foxp3^{Cre}$ mice compared to WT mice (Supplementary Fig. 10). Analysis of the TCR repertoire using RNA-seq reads mapping to the CDR3 region at *Tcrb* showed greater clonal expansion in both Treg cells and CD4+Foxp3− cells isolated from $Tet2/3^{fl/fl}Foxp3^{Cre}$ compared to WT mice (Supplementary Fig. 11 and Supplementary Data 4 and 5), suggesting antigen-driven expansion.

Compared to WT mice, expression of genes involved in Tfh differentiation, including *Gzmb*, *Pdcd1*, *Bcl6*, *Maf*, *Cxcr5*, and *Il21*, was unaltered or only mildly increased in Treg cells isolated from the $Tet2/3^{fl/fl}Foxp3^{Cre}$ mouse with moderate splenomegaly (DKO), but markedly increased in Treg cells isolated from the littermate mouse with more severe splenomegaly (DKO*) (Fig. 4e, left panel). Genes related to Tfh and Th17 cell differentiation (Il17a, Il17f, Rorc) in CD4+Foxp3− T cells displayed a similar trend (Fig. 4e, right panel), indicating that both CD4+Foxp3+ Treg cells and CD4+Foxp3− T cells from $Tet2/3^{fl/fl}Foxp3^{Cre}$ mice displayed skewing to Tfh and/or Th17 lineages in a manner that correlated directly with disease severity. The skewed Tfh and/or Th17 lineages were also observed in Treg cells and CD4+Foxp3− T cells isolated from mLN of $Tet2/3^{fl/fl}Foxp3^{Cre}$ mice (Supplementary Fig. 8e).

In WT mice at steady state, fewer than 5% of CD4+TCRβ+ T cells from spleen and mLNs stained positive for the Tfh signature molecules CXCR5 and PD-1, but this percentage was

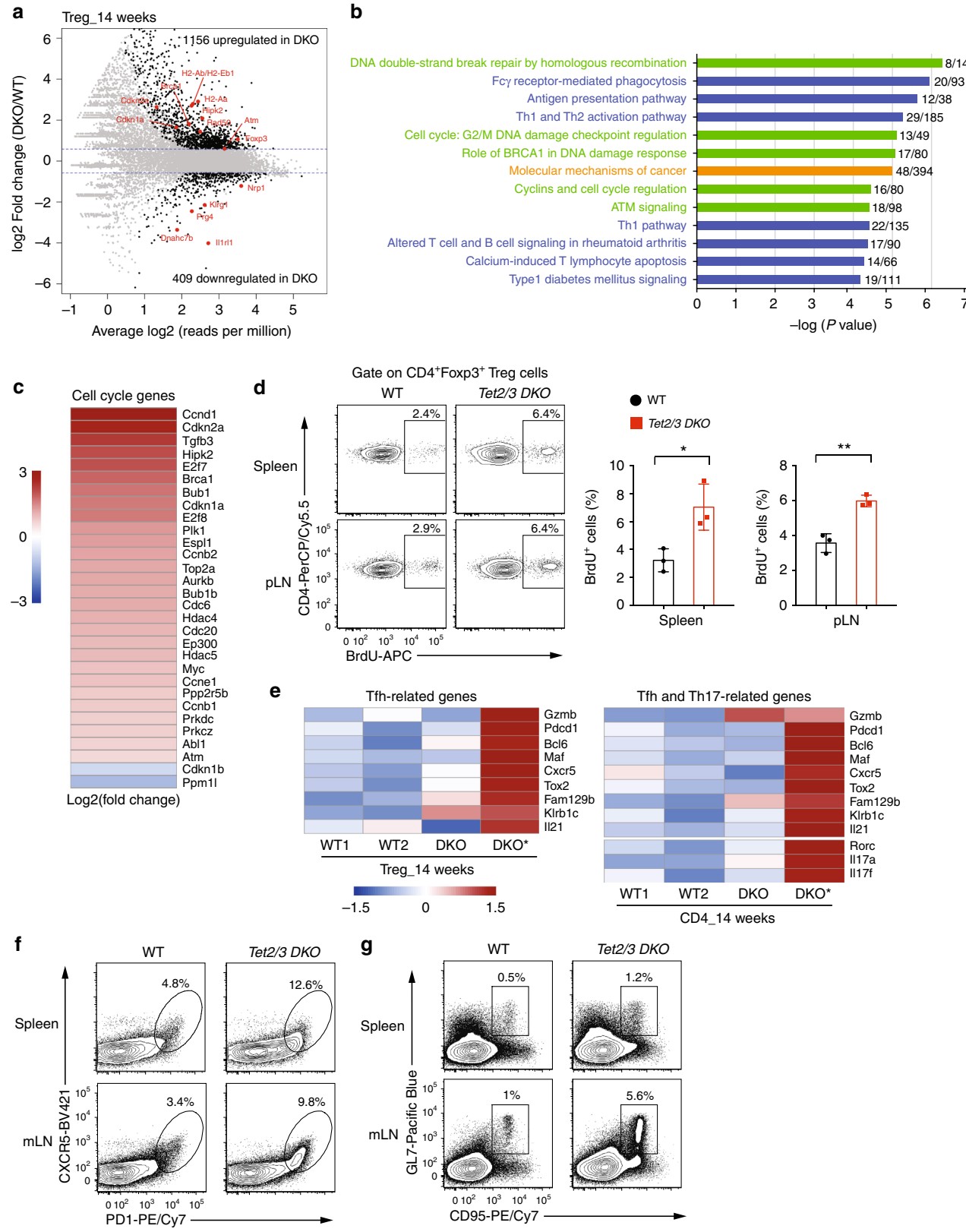

**Fig. 4** (continued across panels a–g)

much higher in the spleen and mLNs of *Tet2/3^fl/fl^Foxp3^Cre^* mice (Fig. 4f). In addition, spleen and mLNs of *Tet2/3^fl/fl^Foxp3^Cre^* mice had more germinal center (GC) B cells, as judged by expression of the GC signature markers GL7 and CD95 (Fig. 4g).

**WT Tregs fail to control autoimmunity in *Tet*-deficient mice.** *Foxp3* is encoded on the X-chromosome, hence due to random X-inactivation, *Tet2/3^fl/fl^* female mice that are heterozygous for *Foxp3^Cre^* (*Foxp3^WT/Cre^*) should harbor both *Tet2/3*-deleted Treg

**Fig. 4** RNA-seq analysis for Treg cells isolated from WT or *Tet2/3*$^{fl/fl}$*Foxp3*$^{Cre}$ DKO mice. **a** Mean average (MA) plot of genes differentially expressed in *Tet2/3* DKO Tregs (14-weeks-old, isolated from pooled spleen and pLNs) relative to their expression in WT Treg cells. **b** IPA analysis of canonical pathways for differentially expressed genes in *Tet2/3* DKO Treg cells. Green, categories related to DNA repair, DNA damage and cell cycle; blue, categories related to immune cell function; orange, category related to cancer. **c** Heatmap for the expression of selected genes encoding cell cycle regulators with differential expression. The color gradient indicates Log2 Fold Change (DKO/WT). **d** *Left panel*, Representative flow cytometry analysis of BrdU incorporation in CD4$^+$Foxp3$^+$ Treg cells in spleen and pLN from WT and *Tet2/3*$^{fl/fl}$*Foxp3*$^{Cre}$ DKO mice (10 weeks old). The gates show the percentage of cells with BrdU incorporation. *Right panel*, Quantification of the frequency of CD4$^+$Foxp3$^+$ Treg cells that incorporated BrdU. **e** Heatmaps showing expression (row z score of log2 TPM values) of Tfh-related genes in WT and DKO Tregs (*left panel*) and of Tfh and Th17 related genes for WT and DKO CD4$^+$Foxp3$^-$ cells (*right panel*). **f**. Flow cytometry analysis of CXCR5$^+$PD-1$^+$ Tfh cells (gated on CD4$^+$TCRβ$^+$ T cells) in the spleen and mLN of WT and *Tet2/3*$^{fl/fl}$*Foxp3*$^{Cre}$ DKO mice (13–16 weeks-old). **g** Flow cytometry analysis of GL7$^+$CD95$^+$ germinal center B cells (gated on CD19$^+$ B cells) in the spleen and mLN of WT and *Tet2/3*$^{fl/fl}$*Foxp3*$^{Cre}$ DKO mice (13–16 weeks old). Statistical analysis was performed using two-tailed unpaired student's *t* test (*$P < 0.05$, **$P < 0.01$, ***$P < 0.001$, ****$P < 0.0001$). Error bars show mean ± s.d. from two to three independent experiments

cells (*Foxp3*$^{Cre}$-positive) and WT Treg cells (*Foxp3*$^{Cre}$-negative). Thus, the presence of WT Treg cells should in theory prevent the mice from developing any autoimmune disease. Surprisingly, we found that *Tet2/3*$^{fl/fl}$*Foxp3*$^{WT/Cre}$ female mice also developed lymphoproliferative disease beginning as early as 15 weeks of age, and all mice succumbed by the age of 38 weeks (Fig. 5a). As expected, the total cellularity of the spleen from *Tet2/3*$^{fl/fl}$ *Foxp3*$^{WT/Cre}$ female mice was significantly increased compared to WT female mice (Fig. 5b).

To confirm this observation independently, we generated mixed bone marrow chimeras by adoptively co-transferring WT (CD45.1$^+$) bone marrow cells together with bone marrow cells (CD45.2$^+$) isolated from WT or *Tet2/3*$^{fl/fl}$*Foxp3*$^{Cre}$ DKO mice at 1:1 ratio into lethally irradiated CD45.1$^+$ WT mice (Fig. 5c). Chimeric mice given either WT or *Tet2/3*$^{fl/fl}$*Foxp3*$^{Cre}$ DKO bone marrow cells were similar and appeared relatively normal at 14–16 weeks after adoptive transfer (Supplementary Fig. 12). However at 18–20 weeks after adoptive transfer, DKO bone marrow chimeras developed splenomegaly compared to WT chimeras and showed a significant increase of total cellularity in the spleen (Fig. 5d).

Notably, CD45.2$^+$ cells in mixed chimeras with *Tet2/3*$^{fl/fl}$ *Foxp3*$^{Cre}$ DKO bone marrow showed considerably higher capacity for expansion compared to CD45.2$^+$ cells in WT mixed bone marrow chimeras (Fig. 5e, f, leftmost panels). In the DKO chimeric mice, the percentage of CD4$^+$CD45.2$^+$ T cells was significantly higher than expected, with over 80% of CD4$^+$CD45.2$^+$ T cells observed in one DKO chimeric mouse; however, the percentage of CD8$^+$ T cells was strongly decreased (Fig. 5e, second panels; Fig. 5f, second and third panels). The frequency of Foxp3$^+$ cells in the CD45.2$^+$CD4$^+$ population varied widely in the DKO mixed chimeras, with some chimeras displaying a frequency as high as 60–70%, while others showed fewer than 10% of CD45.2$^+$CD4$^+$Foxp3$^+$ cells (Fig. 5e, middle panel; Fig. 5f, last panel), suggesting that Treg cells in the DKO chimeras could manifest unstable Foxp3 expression. Moreover, both CD4$^+$Foxp3$^+$ and CD4$^+$Foxp3$^-$ DKO cells in the chimeric mice showed an activated/memory phenotype, with a strong decrease in the proportion of "naïve" CD62L$^{hi}$CD44$^{lo}$ cells (Fig. 5e, last two panels). Together these data indicate that the loss of T regulatory function in *Tet2/3* DKO Treg cells is dominant, and cannot be rescued or prevented in the presence of WT Treg cells.

**Unstable Foxp3 expression in *Tet2/3* DKO Treg cells**. Although Foxp3$^+$ Treg cells are generally stable and are marked by sustained expression of Foxp3, they may lose Foxp3 expression under certain conditions and develop into cells resembling effector T cells, designated "ex-Treg" cells[5]. To examine whether *Tet2/3*-deficient Treg cells indeed lose Foxp3 expression and

become ex-Treg effector cells, we introduced the *Rosa26*-YFP$^{LSL}$ allele[43] into *Tet2/3*$^{fl/fl}$*Foxp3*$^{Cre}$ mice. In the LSL (*LoxP*-STOP-*LoxP*) cassette, a strong transcriptional stop site is flanked by two *LoxP* sites and inserted into the *Rosa26* locus ahead of the cDNA sequence encoding YFP, ensuring that YFP expression is turned on only in Cre-expressing cells. Since YFP expression from the *Rosa26* locus is much brighter than YFP expression from the YFP-Cre fusion protein introduced into the 3' UTR of the *Foxp3* gene, cells expressing YFP from the *Rosa26* locus can be easily distinguished from cells expressing the YFP-Cre fusion protein encoded in the *Foxp3* gene[44] (Fig. 6a, where intermediate YFP expression derives from the YFP-Cre fusion protein whereas high YFP expression derives from the *Rosa26* fate mapping allele). Analysis of CD4$^+$ T cells from spleen and mLNs of *Tet2/3*$^{fl/fl}$ *Foxp3*$^{Cre}$*Rosa26*-YFP$^{LSL}$ mice showed clearly that a much higher proportion of Rosa-YFP$^{high}$ cells—cells that had turned on Cre from the *Foxp3* gene, and thus had been Treg cells earlier in their developmental history—had lost Foxp3 expression and turned into ex-Treg cells, compared to Rosa-YFP$^{high}$ cells from heterozygous *Tet2*$^{+/fl}$*Tet3*$^{fl/fl}$*Foxp3*$^{Cre}$*Rosa26*-YFP mice (Fig. 6a, b). Further confirming the existence of ex-Treg cells in *Tet2/3*$^{fl/fl}$ *Foxp3*$^{Cre}$ mice, we showed by quantitative real-time PCR, as well as RNA-seq that *Tet2* and *Tet3* mRNAs were both greatly reduced in expression in CD4$^+$YFP$^-$ cells isolated from *Tet2/3*$^{fl/fl}$*Foxp3*$^{Cre}$ compared to WT mice (qPCR, Supplementary Fig. 13a; number of normalized reads mapping to the deleted exons; Supplementary Fig. 13b–c). Taken together with the data presented in the previous sections, this finding suggests strongly that a substantial fraction of the activated CD4$^+$YFP$^-$ T cells in *Tet2/3*$^{fl/fl}$*Foxp3*$^{Cre}$ mice (Fig. 5e, f) are ex-Treg cells that have lost suppressive function and acquired deleterious effector function.

***Tet2/3* DKO CD4$^+$ T cells elicits disease in healthy mice**. To test the above hypothesis of deleterious effector function acquired by ex-Treg cells from *Tet2/3*$^{fl/fl}$*Foxp3*$^{Cre}$ mice, we asked whether total CD4$^+$ T cells from *Tet2/3*$^{fl/fl}$*Foxp3*$^{Cre}$ mice, which contain ex-Treg cells, as well as bystander activated CD4$^+$ T cells, could elicit disease when transferred into healthy immunocompetent recipient mice. We transferred purified CD4$^+$ T cells ($5 \times 10^6$; purity > 98.5%) from spleen and pLNs into fully immunocompetent (non-irradiated) congenic CD45.1$^+$ recipient mice (Fig. 7a). All the recipients developed enlarged spleen and lymph nodes and developed disease 7–10 weeks after transfer (Fig. 7b). *Tet2/3* DKO CD4$^+$ T cells expanded 3–50 fold, from the starting transferred population of 5 million cells to as many as 250 million cells (Fig. 7b). The percentage of Foxp3$^+$ cells after expansion of the transferred CD4$^+$ population varied from mouse to mouse, with three mice showing high-level Foxp3 expression (mouse example 1 in Fig. 7c), and the others showing low to no expression of Foxp3 (mouse example 2 in Fig. 7c). We observed a compensatory

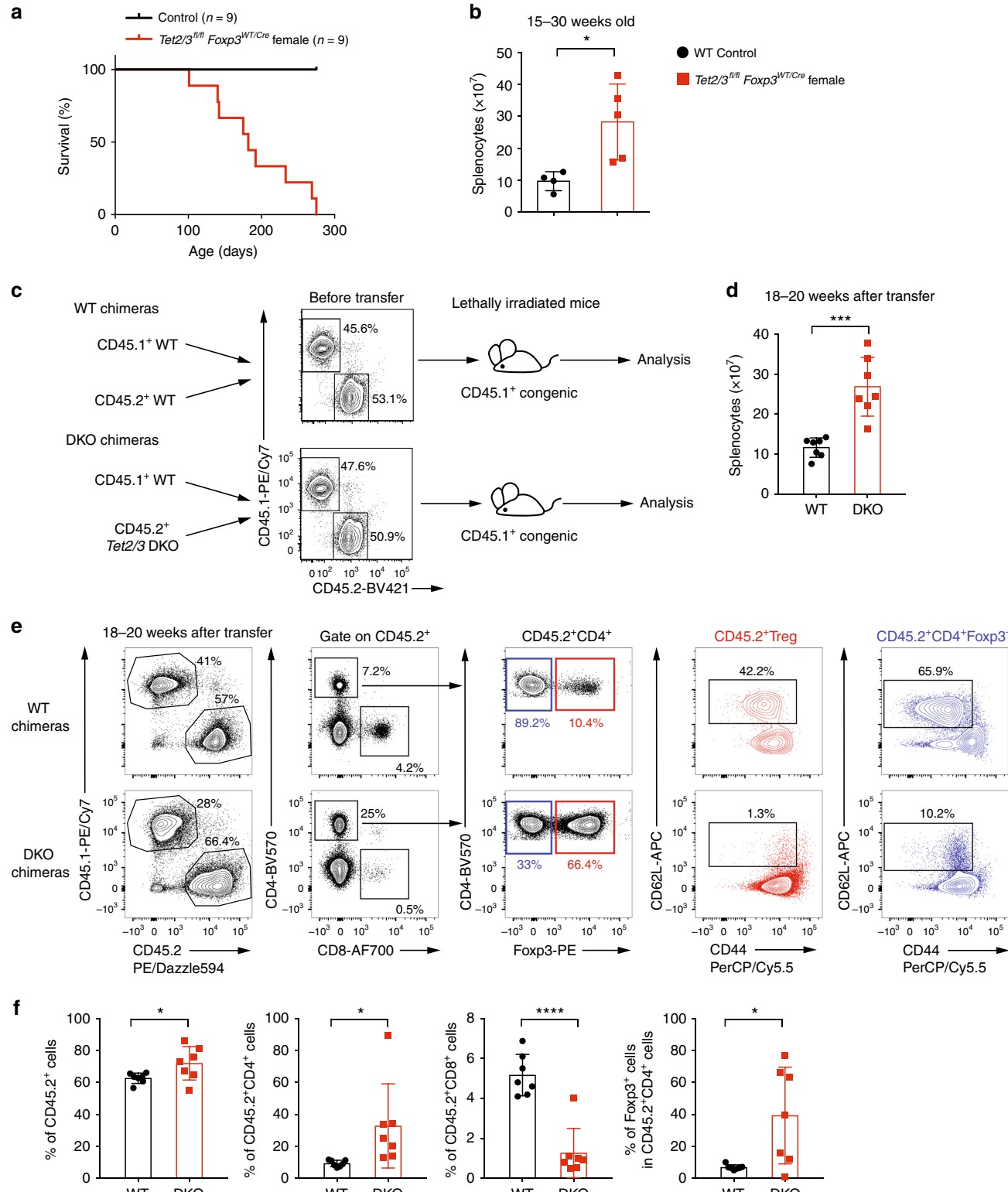

**Fig. 5** WT Treg cells fails to rescue the inflammatory phenotype of $Tet2/3^{fl/fl}Foxp3^{Cre}$ mice. **a** Survival curves for control WT ($n = 9$) and $Tet2/3^{fl/fl}$ $Foxp3^{WT/Cre}$ heterozygous female ($n = 9$) mice. **b** Total cell numbers in spleen from WT and $Tet2/3^{fl/fl}Foxp3^{WT/Cre}$ heterozygous female mice (15–30 weeks-old, $n = 5$). **c** Schematic illustration for the generation of mixed bone marrow chimeras. The flow cytometry plots show the ratios of WT CD45.1$^+$ congenic bone marrow cells and co-transferred CD45.2$^+$ WT or $Tet2/3$ DKO bone marrow cells immediately prior to transfer. **d** Total cell numbers in spleen from WT and $Tet2/3$ DKO mixed bone marrow chimeras with splenomegaly ($n = 7$) 18–20 weeks after transfer. **e** Representative flow cytometry analysis for WT and DKO mixed bone marrow chimeras 18–20 weeks after transfer. **f** The graphs from left to right show the quantifications for the percentage of CD45.2$^+$ cells, the percentage of CD4$^+$ and CD8$^+$ cells within the CD45.2$^+$ cells and the percentage of Foxp3$^+$ cells within the CD45.2$^+$CD4$^+$ cells ($n = 7$). Statistical analysis was performed using two-tailed unpaired student's t test (*$P < 0.05$, **$P < 0.01$, ***$P < 0.001$, ****$P < 0.0001$). Error bars show mean ± s.d. from three independent experiments

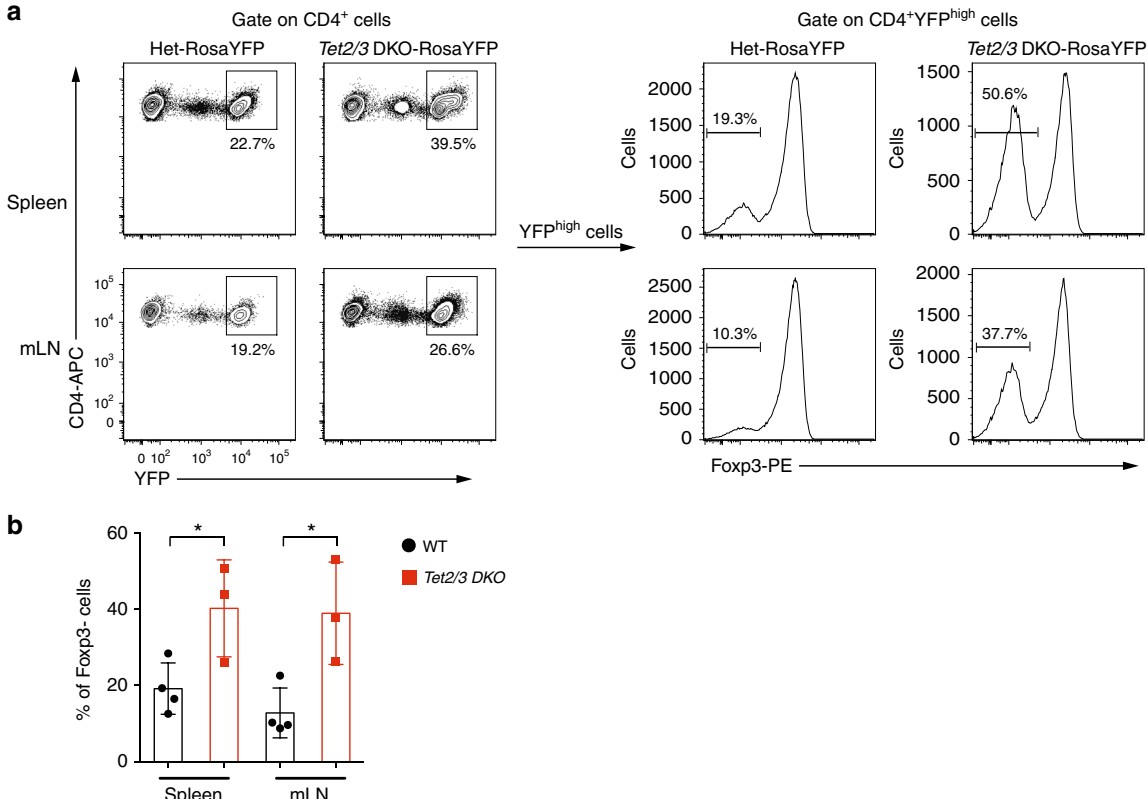

**Fig. 6** *Tet2/3 DKO* Treg cells are less stable than Treg cells from heterozygous littermates. **a** Representative lineage tracing of Treg cells in spleen and mLN from littermate mice of *Tet2+/flTet3fl/flFoxp3YFP-CreRosa26YFP* (Het) and *Tet2/3fl/flFoxp3YFP-CreRosa26YFP* genotypes. *Left panels*, Contour plots showing that Rosa26YFP is much brighter than, and so can be distinguished from, Foxp3YFP-Cre. *Right panels*, The percentage of Foxp3-negative cells is much higher in sorted CD4+Rosa26YFPhigh cells from *Tet2/3fl/flFoxp3YFP-CreRosa26YFP* (*Tet2/3 DKO*) mice compared to those from *Tet2+/flTet3fl/flFoxp3YFP-CreRosa26YFP* mice, indicating that *Tet2/3 DKO* Treg cells that had originally turned on Foxp3 and Cre from the *Foxp3YFP-Cre* locus later lose Foxp3 and become ex-Treg cells. **b** Quantification of the percentage of Foxp3neg-Rosa26YFPhigh cells as shown in **a** (11–12 weeks old, n = 3 from three independent experiments). Statistical analysis was performed using two-tailed unpaired student's t test (*P < 0.05). Error bars show mean ± s.d. from three independent experiments

expression of Foxp3+ cells in CD45.1+ host cells in each individual (Fig. 7b, c). Moreover, both CD4+Foxp3+ host Treg cells and CD4+Foxp3− host cells displayed activated phenotype with low levels of CD62L expression and high levels of CD44 expression (Fig. 7c, Host Cells). Together these data suggest that *Tet2/3 DKO* CD4+ T cells contained the cell population responsible for transferring the diseases to immunocompetent mice. These might be ex-Treg effector cells, host CD4+ T cells that became activated and converted to effector cells, or both.

## Discussion

Here, we investigated the role of two epigenetic regulators, the methylcytosine oxidases Tet2 and Tet3, in Treg function. Mice with Treg-specific deletion of *Tet2* and *Tet3* (*Tet2/3fl/flFoxp3Cre* mice) display defective Treg function and develop an inflammatory disease characterized by splenomegaly and leukocyte infiltration into tissues. TET loss-of-function in developing *Tet2/3fl/flFoxp3Cre* Treg cells led to increased methylation (5mC + 5hmC) at the *Foxp3 CNS1* and *CNS2* enhancers compared to wild-type Treg cells resulting in decreased stability of Foxp3 expression and impaired Treg cell function. Notably, both ex-Treg cells and conventional CD4+ and CD8+ T cells in *Tet2/3fl/flFoxp3Cre DKO* mice developed an aberrant dominant effector phenotype, which could not be suppressed by the presence of wild-type Treg cells in *Tet2/3fl/flFoxp3WT/Cre* heterozygous females or in mixed bone marrow chimeras. Our data emphasize the essential role of TET-mediated epigenetic modifications not only in maintaining the demethylated status of the *Foxp3 CNS1* and *CNS2* enhancers—

thereby stabilizing Foxp3 expression, specifying Treg cell lineage and conferring stable Treg identity—but also in preventing indirectly the acquisition of aberrant effector function by bystander CD4+ and CD8+ T cells with otherwise normal TET function.

Individual or dual deletion of TET genes can both be deleterious for Treg cell[12,26,45] (and this study), B cell[23] and myeloid function[22,46–51]. The phenotypic variability is potentially related to the animal facility in which the mice are housed[41]. Myeloid expansion in *Tet2*-deficient mice has been attributed to the influence of cytokines such as IL-6 produced in response to gut microbiota, since it is eliminated or greatly attenuated if the mice are housed in a germ-free facility[41]. Similarly, the large increase in the cellularity of mesenteric lymph nodes compared to peripheral lymph nodes in our 12–16 week-old *Tet2/3fl/flFoxp3Cre* mice, as well as the increased clonality of both Foxp3+ Treg cells and CD4+Foxp3− T cells (ex-Treg cells and/or activated CD4+ bystander T cells), suggests a role for recognition of microbial or other antigens. Thus, the effects of TET loss-of-function appear most striking when TET-deficient cells are subjected to stimulation, through antigen recognition[25], cytokine stimulation[11], or the influence of the microbiota as discussed above[41].

By fate-mapping experiments, Foxp3 expression in Treg cells from *Tet2/3fl/flFoxp3Cre* mice was considerably less stable than Foxp3 expression in Treg cells from control mice (Fig. 6). Treg cells from *Tet2/3fl/flFoxp3Cre* mice were more prone to lose Foxp3 expression, and also showed dysregulated expression of many Treg signature genes including *Cd25*, *Nrp1*, and *Il1rl1*. Notably, the

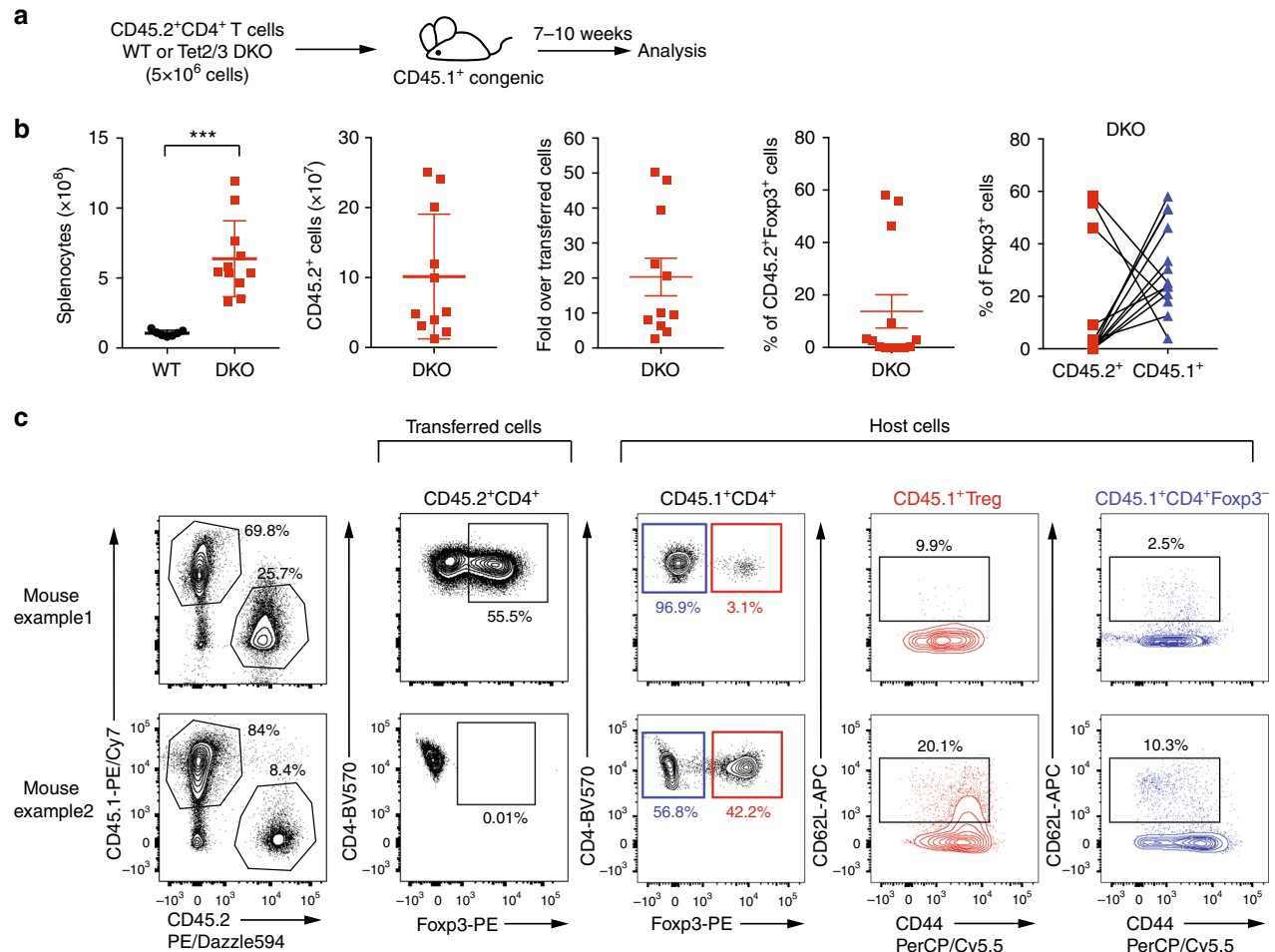

**Fig. 7** Transfer of total CD4$^+$ T cells from *Tet2/3*$^{fl/fl}$*Foxp3*$^{Cre}$ mice elicits disease in healthy mice. **a** Schematic illustration for the transfer of total CD4$^+$ T cells (5 × 10$^6$ cells) from WT or *Tet2/3*$^{fl/fl}$*Foxp3*$^{Cre}$ DKO mice (14 weeks old) into immunocompetent congenic recipient mice (*n* = 11 recipient mice), the mice were analyzed 7–10 weeks after transfer. **b** From left to right, the graphs show the numbers of total splenocytes and of transferred and expanded CD45.2$^+$ cells, the fold increase in the number of transferred and expanded DKO CD45.2$^+$ cells, the percentage of Foxp3$^+$ cells in CD45.2$^+$ cells after transfer, and the percentage of Foxp3$^+$ cells in CD45.2$^+$ and CD45.1$^+$ cells in each individual with connection lines. **c** Representative flow cytometry analyses of the percentage of CD45.2$^+$ T cells, the percentage of Foxp3$^+$ cells in CD45.2$^+$ cells (example 1 showing an expansion of Foxp3$^+$ cells, example 2 showing a loss of Foxp3$^+$ cells), the percentage of Foxp3$^+$ cells in CD45.1$^+$ host cells, CD62L and CD44 expression in host Treg cells and CD4$^+$Foxp3$^-$ cells. Statistical analysis was performed using two-tailed unpaired student's *t* test (***$P$ < 0.001). Error bars show mean ± s.d. from three independent experiments

presence of wild-type Treg cells in *Tet2/3*$^{fl/fl}$*Foxp3*$^{WT/Cre}$ heterozygous females or mixed bone marrow chimeras was not sufficient to suppress the aberrant inflammation observed in *Tet2/3*$^{fl/fl}$*Foxp3*$^{Cre}$ mice, moreover, transfer of total CD4$^+$ T cells from *Tet2/3*$^{fl/fl}$*Foxp3*$^{Cre}$ mice into immunocompetent recipient mice unexpectedly resulted in efficient transfer of the inflammatory disease. In both mixed bone marrow chimeras and CD4$^+$ T cell transfer experiments, Foxp3$^+$ cells from *Tet2/3*$^{fl/fl}$*Foxp3*$^{Cre}$ mice expanded somewhat more extensively than control cells in an early phase in which they maintained almost normal suppressive function; later, they lost Foxp3 expression and suppressive function but acquired effector function, leading to the accumulation of effector ex-Treg cells, as well as activated bystander CD4$^+$ and CD8$^+$ T cells as the mice aged. Thus, loss of TET function in Treg cells of *Tet2/3*$^{fl/fl}$*Foxp3*$^{Cre}$ mice results in a paradoxical dominant disease, in which defective TET-deficient Treg cells lose Foxp3 expression and suppressor function on the one hand, and convert to ex-Treg cells with dominant effector function on the other hand.

In addition to simply losing suppressive function, ex-Treg cells appear to be prone to acquire effector function[52]. The underlying mechanisms are not yet completely clear, but it is known that mice with Treg perturbations frequently compensate for insufficiency of T regulatory function by becoming hyperproliferative and hyperactivated[8,35]. For instance, ex-Treg cells arising as a result of deficiency in the mTOR-signaling regulator TSC1 produce increased levels of IL-17 and acquire Th17 effector features under inflammatory conditions;[53] and neuropilin1-deficient Treg cells produce high levels of IFNγ, which feeds back to suppress Treg cell function[11]. In our case, we have shown that *Tet2/3* deficiency leads to increased expression of genes related to Tfh and/or Th17 cell differentiation, both in CD4$^+$Foxp3$^-$ T cells which include both ex-Treg cells and CD4$^+$ bystander cells, and in CD4$^+$Foxp3$^+$ Treg cells, in a manner that correlated directly with disease severity.

Our data emphasize the critical role of TET proteins in the maintenance of stable Foxp3 expression and the integrity of Treg gene expression and function. Absence of TET proteins in Treg cells sets off a cascade of events that begins with increased DNA methylation of *CNS1* and *CNS2* and unstable Foxp3 expression, both facilitated by cell division occurring as a result of antigen

and cytokine stimulation; and continues into compromised suppressor function and acquisition of effector function by ex-Treg cells, as well as bystander T cells. The aberrant acquisition of these dysregulated phenotypes eventually results in a fatal inflammatory/ lymphoproliferative disease.

## Methods

**Mice**. B6.129(Cg)-$Foxp3^{tm4(YFP/icre)Ayr}$/J ($Foxp3^{Cre}$, strain 016959), B6.SJL-Ptprc$^a$Pepc$^b$/BoyJ (CD45.1 congenic mice, strain 002014), B6.129S7-$Rag1^{tm1Mom}$/J ($Rag1^{-/-}$ mice, strain 002216), and B6.Cg-$Foxp3^{sf}$/J (B6-scurfy mice, strain 004088) were obtained from Jackson Laboratory. $Tet2^{fl/fl}Tet3^{fl/fl}$Foxp3$^{Cre}$ mice were generated in our laboratory by crossing $Tet2^{fl/fl}Tet3^{fl/f}$ with $Foxp3^{Cre}$ mice. For the fate-mapping experiments, $Tet2^{+/fl}Tet3^{fl/fl}Foxp3^{Cre}$ or $Tet2^{fl/fl}Tet3^{+/fl}Foxp3^{Cre}$ female mice were crossed with $Tet2^{fl/fl}Tet3^{fl/fl}Rosa26$-YFP$^+$ male mice to generate $Tet2^{fl/fl}Tet3^{fl/fl}Foxp3^{Cre}Rosa26$-YFP$^+$ male mice for further analysis. All mice were on the B6 background and maintained in a specific pathogen-free animal facility in the La Jolla Institute for Immunology. Age of the mice used for each experiment was stated in the figure legends. All cell or mouse irradiation procedures were performed using RS2000 Biological Irradiator (Rad Source Technologies, Inc.) All animal procedures were reviewed and approved by the Institutional Animal Care and Use Committee of the La Jolla Institute for Immunology and were conducted in accordance with institutional guidelines.

**Cell preparation and flow cytometry**. Single-cell suspensions were prepared from spleen, peripheral lymph nodes, and mesenteric lymph nodes for staining or cell sorting. For analysis of T cell compartments and Treg cell features, single-cell suspensions were stained with anti-mouse antibodies against the following (Clone name, conjugated fluorescence, dilution, manufacturer and catalog number shown in brackets): CD4 (RM4-5, PerCP-Cy5.5, 1:200, Biolegend, #100540; GK1.5, APC, 1:200, Biolegend, #100412; RM4-5, BV570, 1:200, Biolegend, #100541; GK1.5, BV421, 1:200, Biolegend, #100437), CD8 (53-6.7, PE-Cy7, 1:200, Biolegend, #100722; 53-6.7, AF700, 1:200, Biolegend, #100730), CD62L (MEL-14, BV421, 1:400, Biolegend, #104436; MEL-14, APC, 1:200, Biolegend, #104412), CD25 (PC61, APC, 1:200, Biolegend, #102012), CD44 (IM7, PE, 1:200, Biolegend, #103008; IM7, PerCP-Cy5.5, 1:200, Biolegend, #103032), CD45.1 (A20, PE-Cy7, 1:200, Biolegend, #110730), CD45.2 (104, PE-Dazzle594, 1:200, Biolegend, #109846; 104, BV421, 1:200, Biolegend, #109831), Nrp1 (3E12, BV421, 1:200, Biolegend, #145209), ICOS (C398.4 A, AF488, 1:200, Biolegend, #313514), CD103 (2E7, FITC, 1:200, Biolegend, #121419), GITR (YGITR765, PE-Cy7, 1:200, Biolegend, #120222), PD1 (29 F.1A12, PE-Cy7, 1:200, Biolegend, #135216), CD127 (A7R34, BV421, 1:200, Biolegend, #135027), CD69 (H1.2F3, APC, 1:200, Biolegend, #104514), B220 (RA3-6B2, APC, 1:200, Biolegend, #103212), Gr1 (RB6-8C5, PE, 1:400, Biolegend, #108408), CD11b (M1/70, PerCP-Cy5.5, 1:200, Biolegend, #101228), CCR7 (4B12, BV421, 1:100, Biolegend, #120119). Anti-mouse CCR7 staining was performed at 37 °C for 30 min. For analysis of Tfh and germinal center B cells, single-cell suspensions were stained with anti-mouse antibodies TCRβ (H57-597, APC, 1:200, Biolegend, #109212), CD19 (6D5, APC, 1:200, Biolegend, #115512), CXCR5 (L138D7, BV421, 1:100, Biolegend, #145511), PD1 (29 F.1A12, PE-Cy7, 1:200, Biolegend, #135216), GL7 (GL7, Pacific Blue, 1:200, Biolegend, #144613) and CD95 (Fas, Jo2, PE-Cy7, 1:200, BD Biosciences, #557653). For intracellular staining, cells were surface-stained and then stained with anti-Foxp3 (FJK-16S, PE, 1:100, eBioscience, #12-5773-82), anti-active caspase-3 (C92-605, PE, 20 µl per assay, BD Biosciences, #550821), anti-Helios (22F6, APC, 1:100, Biolegend, #137221), anti-CTLA4 (UC104.89, APC, 1:100, Biolegend, #106309), and anti-H2A.X phosphorylated (Ser139, 2F3, APC, 1:100, Biolegend, #613415) antibodies using the Foxp3 Fixation/Permeabilization kit (eBioscience, #00-5523-00) and analyzed by flow cytometry on LSR-II and LSR Fortessa.

**BrdU incorporation assay**. Mice were injected with 1 mg of BrdU (100 µl of 100 mg/ml stock solution) intraperitoneally. 24 h later, mice were sacrificed and the cells from spleen and peripheral lymph nodes were stained for surface markers and then for BrdU according to the manufacturer's protocol (BD Pharmingen, APC BrdU flow kit 552598).

**Anti-dsDNA antibody ELISA**. Anti-dsDNA autoantibodies in the serum from $Foxp3^{Cre}$ WT and $Tet2/3^{fl/fl}Foxp3^{Cre}$ mice (11–16 weeks-old) were measured with kit from Alpha Diagnostic International according to the manufacturer's protocol (ELISA Kit 5110).

**LEGENDplex mouse immunoglobulin isotyping panel**. The serum immunoglobulin levels were measured with LEGENDplex Mouse Immunoglobulin Isotyping Panel according to the manufacturer's protocol (Biolegend, 740493).

**Analysis of peripheral bloods from $Tet2/3^{fl/fl}$Foxp3$^{Cre}$ mice**. Peripheral bloods from WT and $Tet2/3^{fl/fl}Foxp3^{Cre}$ DKO mice (12–14 weeks-old) were collected into tubes pretreated with EDTA (BD Biosciences, 365974). Hematological parameters including the concentrations of white blood cells (WBCs), red blood cells (RBCs), and platelets were analyzed by Hemavet 950FS (Drew Scientific).

**Generation of mixed bone marrow chimeras**. Bone marrow cells were obtained from tibia and fibula from $Foxp3^{Cre}$ WT mice, $Tet2/3^{fl/fl}Foxp3^{Cre}$ mice and CD45.1 congenic mice. Erythrocytes and mature T cells were then depleted from the bone marrow cells using MACS LS columns. The purified cells (2.5 × 10$^6$) from $Foxp3^{Cre}$ WT and $Tet2/3^{fl/fl}Foxp3^{Cre}$ mice were then mixed together with cells (2.5 × 10$^6$) isolated from tibia and fibula from CD45.1 congenic mice at 1:1 ratio and transferred intravenously (i.v.) into lethally irradiated CD45.1 congenic mice. Reconstituted mice were then sacrificed for analysis 14–20 weeks after adoptive transfer of bone marrow cells.

**$Scurfy$ CD4$^+$ T cell adoptive transfer**. $Scurfy$ CD4$^+$ T cells were isolated from spleen and peripheral lymph nodes from male $Scurfy$ mice and purified using Dynabeads (Life Tech, purity > 98%). 5 × 10$^5$ CD45.1$^+$CD4$^+$ $scurfy$ T cells were injected into $Rag1$-deficient mice alone or mixed with 1 × 10$^5$ Treg cells isolated from $Foxp3^{Cre}$ WT or $Tet2/3^{fl/fl}Foxp3^{Cre}$ mice. Four to 5 weeks after adoptive transfer, the percentage of CD45.1$^+$CD4$^+$ cells and CD45.2$^+$ CD4$^+$Foxp3$^+$ cells were analyzed by flow cytometry. Alternatively, we used bone marrow cells from scurfy mice, $Foxp3^{Cre}$ WT or $Tet2/3^{fl/fl}Foxp3^{Cre}$ mice for the adoptive transfer.

**Tet2/3 DKO CD4$^+$ T cell adoptive transfer**. Total CD4$^+$ T cells were isolated from spleen and peripheral lymph nodes from CD45.2$^+$ $Tet2/3^{fl/fl}Foxp3^{Cre}$ mice and double purified using Dynabeads (Life Tech, purity > 98.5%). 5 × 10$^6$ CD4$^+$ T cells were adoptively transferred into immunocompetent congenic mice and analyzed for 7–10 weeks after adoptive transfer. As a control, total CD4$^+$ T cells isolated from CD45.2$^+$ $Foxp3^{Cre}$ WT mice were also transferred into immunocompetent congenic mice, which did not show any cell expansion.

**Bisulfite (BS) sequencing**. DNA samples were treated with sodium bisulfite for 4 h (MethylCode Bisulfite Conversion Kit, MECOV50, Invitrogen). The PCR amplicons were generated using PyroMark PCR kit (Qiagen), and quantified using Quant-iT PicoGreen dsDNA reagent (Invitrogen). PCR amplicons were then used for library preparation using NEBNext DNA Library Modules for Illumina platform (NEB). The final libraries were quantified using KAPA library quantification kit for Illumina (KAPA Biosystems), and sequenced on Miseq (300 bp, paired end, Illumina). The data are based on thousands of sequence reads per amplicon. The bisulfite sequencing reads were mapped to mouse genome mm9 using the Bismark mapping tool. The mapping was done using the paired-end Bowtie2 backend with the following parameter values: -I 0 –X 600 –N 0. For each of the samples, the "bismark_methylation_extractor" script in the Bismark package was used to extract the numbers of times each cytosine within the amplicons was converted. These counts were used to calculate the proportions for each cytosine to be non-methylated (or formylmethylated/carboxylmethylated) or methylated (or hydroxymethylated).

**RNA-seq library preparation**. Total RNA were isolated from CD4$^+$YFP$^+$ (Foxp3$^+$) Treg cells and CD4$^+$YFP$^-$ (Foxp3$^-$) T cells (from pooled spleen and pLNs in Fig. 4 or from mLNs in Supplementary Fig. 8) from $Foxp3^{Cre}$ WT mice and $Tet2/3^{fl/fl}Foxp3^{Cre}$ DKO mice (14-weeks-old) using RNeasy plus mini kit (Qiagen). RNA-sequencing libraries were prepared using Truseq stranded mRNA kit (Illumina) according to the manufacture's protocol and sequenced at the La Jolla Institute sequencing core using Illumina HiSeq2500 single end 50 bp platform.

**RNA-seq and TCR-seq analysis**. RNA-seq data were mapped against the UCSC mouse genome mm9 using TopHat[54] (v2.1.1) with the following parameters "-p 16 -N 2 --max-multihits 1 --read-gap-length 1 --transcriptome-index" and the $RefSeq$ gene annotation was obtained from the UCSC genome Bioinformatics database. The number of reads mapping to each gene was counted using featureCounts[55] (subread-1.4.3-p1) with the following parameters "-g gene_name –s 2". Differentially expressed genes (DEGs) between WT and Tet2/3 DKO cell types were determined using the Bioconductor package DESeq2[56] with adjusted $P$ value < 0.05 and a fold change threshold of > 1.5 or < 0.67. And genes with total counts < 1 in the sum of all conditions were removed from the analysis. Canonical pathway analysis was performed using Ingenuity Pathway Analysis software (license for La Jolla Institute).

TCR sequences were retrieved from RNA-Seq data sets, and the frequency of TCRβ chain clonotypes (CDR3 regions) was determined using MiXCR[57] (mixcr-1.7–2.1) package for RNA-seq analysis with the default parameters "align –l TCR –s mmu –p rnaseq -OallowPartialAlignments = true". Two rounds of contig assembly were performed by employing the "assemblePartial" function; extension of incomplete TCR CDR3s with uniquely determined V and J genes using germline sequences was done using the "extendAlignments" function; assembly and export of the clonotypes was performed using the "assemble" and the "exportClones" (--preset min -fraction -targets -vHits -dHits -jHits -vAlignments -dAlignments -jAlignments) functions, respectively.

**Quantitative real-time PCR**. Total RNA was isolated using RNeasy plus mini kit (Qiagen); cDNA was synthesized using SuperScript III reverse transcriptase (Thermo Fisher). Quantitative real-time PCR was performed using FastStart Universal SYBR Green Master mix (Roche) on a StepOnePlus real-time PCR machine (Thermo Fisher). Gene expression was normalized to *Hprt*. Primers used to detect the expression levels of *Tet2*, *Tet3*, and *Foxp3* are as following:

*Tet2* forward primer: AACCTGGCTACTGTCATTGCTCCA
*Tet2* reverse primer: ATGTTCTGCTGGTCTCTGTGGGAA
*Tet3* forward primer: GTCTCCCCAGTCCTACCTCCG
*Tet3* reverse primer: GTCAGTGCCCCACGCTTCA
*Foxp3* forward primer: GGCCCTTCTCCAGGACAGA
*Foxp3* reverse primer: GCTGATCATGGCTGGGTTGT
*Hprt* forward primer: CTGGTGAAAAGGACCTCTCG
*Hprt* reverse primer: TGAAGTACTCATTATAGTCAAGGGCA

**Histology**. Lungs, livers, and spleens were isolated from 12–14-week-old *Tet2/3*[fl/fl]*Foxp3*[Cre] DKO mice and littermate controls. Samples were fixed using 10% formalin overnight, preserved in 70% ethanol and embedded in paraffin. The paraffin blocks were then cut and sections were stained with haematoxylin and eosin (H&E).

**Statistics**. P values from unpaired two-tailed Student's *t* test were used for all the statistical comparisons between different groups and data were displayed as mean ± s.d. (Prism). P values are denoted in corresponding figures as: *$P < 0.05$, **$P < 0.01$, ***$P < 0.001$, ****$P < 0.0001$.

**Reporting summary**. Further information on experimental design is available in the Nature Research Reporting Summary linked to this article.

## Data availability

The authors declare that all data supporting the findings of this study are available within the article and its Supplementary information files or from the corresponding author upon reasonable request. RNA-seq data have been deposited in the Gene Expression Omnibus database under accession number GSE113694. A reporting summary for this Article is available as a Supplementary Information file.

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

## Acknowledgements
We thank members of the Rao laboratory for suggestions and discussions. We thank Cheryl Kim, Denise Hinz, Lara Boggeman, and Robin Simmons at the La Jolla Institute Flow Cytometry facility for help with cell sorting experiments; Jeremy Day of the La Jolla Institute Sequencing facility for help with next-generation sequencing and Dr. Zbigniew Mikulski and Margie Chadwell of the La Jolla Institute Microscopy and Histology facility for help with histological and microscopic analysis; and the Histology Core at the University of California at San Diego Moores Cancer Center. We thank Dr. Nissi Varki from University of California at San Diego for help with histology analysis. This work was supported by National Institutes of Health (NIH) R01 grants R35 CA210043 and AI 12858901 (to A.R.). FACSAria II Cell Sorter was acquired through the Shared Instru- mentation Grant (SIG) Program S10 RR027366 and Hiseq2500 was funded by S10OD016262. C.W.L. was supported by Irvington Postdoctoral Fellowship from the Cancer Research Institute. D.S.C. is a graduate student in the UCSD Biology Program and is supported by a CONACYT/UCMEXUS fellowship from Mexico. X.L. was sup- ported by a postdoctoral Fellowship from CIRM UCSD Interdisciplinary Stem Cell Research and Training Grant II (TG2-01154).

## Author contributions
A.R., X.Y., and C.W.L. conceived the project. A.R. supervised project planning and execution. X.Y. and X.L. performed experiments. D.S.C. performed bioinformatic ana- lyses of RNA-seq data. C.W.L. provided input for experimental design and critically evaluated the manuscript. A.R. and X.Y. wrote the manuscript.

## Additional information

**Competing interests:** A.R. is a member of the Scientific Advisory Board of Cambridge EpiGenetix, Ltd., Cambridge, UK. The remaining authors declare no competing interests.

**Reprints and permission** information is available online at http://npg.nature.com/ reprintsandpermissions/

