## [Peer Review File · Nature Communications]

Reviewers' comments:

Reviewer #1 (Remarks to the Author):

This paper by Yue et al. analyzed the role of Tet2/Tet3 in Tregs by generating Treg-specific Tet2/3-double deficient (DKO) mice. They found that DKO mice develop lymphoproliferative and inflammatory diseases. Both Foxp3⁺ and Foxp3⁻ cells showed strong skewing to Tfh and/or Th17 phenotypes. Treg cells from DKO mice were more prone to lose Foxp3 expression and lost suppression activity in vivo. Authors concluded that Tet2 and Tet3 are guardians of Treg cell stability and immune homeostasis in mice.

This paper extends their previous reports showing unstable Foxp3 expression in Tet2/Tet3 DKO Tregs (Yue et al. J. Exp. Med. 2016). In addition, they discovered a strong phenotype of Th17/Tfh skewing and fatal lymphoproliferative disease in their mice. They discovered altered expression of Th17/Tfh genes, cell cycle related genes and DNA damage repair genes in Tregs in their DKO mice. These phenotypes are very interesting, even though the mechanism is still an open question. I have several suggestions which may improve the value of the paper.

(1) Previously authors showed that TCR-dependent iNKT cell expansion in T cell-specific (CD4Cre)-DKO mice. Likewise Foxp3⁺Tregs were expanded in their Foxp3Cre-DKO mice. In Supplementary Fig.4, authors showed RNA-seq data of CDR3 suggesting a clonal expansion in both Treg cells and CD4⁺Foxp3⁻ cells. It is not very clear whether expansion of Foxp3⁻ cells is due to "bystander expansion" or "expansion of exFoxp3 cells". Therefore, it may be better if authors can show the actual sequences of CDR3 from DKO Foxp3⁺ and Foxp3⁻ cells. In addition, it is informative to check whether expanded Foxp3⁻ cells lacked Tet2/Tet3 genes or not. If expansion of Foxp3⁻ cells is a bystander effect, how do authors explain this? High levels of IL-2?

(2) Similarly, Foxp3⁻ cells tend to become Th17/Tfh-like cells due to high levels of IL-6 or IL-21? This can be addressed in vitro culture of Tregs with these cytokines and measurement of Th17/Tfh phenotypes.

(3) Previous paper using CD4Cre-DKO indicates that TET-mediated CNS2 demethylation is necessary for maintenance of iTregs in vivo. Stable iTregs after transfer showed CNS2 demethylation in a TET-dependent manner. Authors should check stability and CNS2 methylation of Foxp3 in iTregs from Foxp3CreDKO mice after transfer. This experiment may define the essential and/or redundant role of TETs in Foxp3 stability in mature Tregs.

(4) Figs.3/4: Unstable Foxp3 expression is probably due to reduced demethylation of CNS1 and CNS2. However, methylation status of other TSDRs or Treg related genes seems not to be examined. Are there any correlations between Foxp3 instability/signature genes expression and their (TSDR) methylation status?

Minor comments

(1) DNA damage stress has been shown to be associated with malignant proliferation of Tets-deficient cells. How about DNA damages (γ -H2AX) in Foxp3⁺ and Foxp3⁻ cells?

(2) Authors mentioned the work by Bana Jabri (Nature, in press), which shows the importance of commensal bacteria for the phenotypes of Tet2-deficient mice. It will be informative if authors examine pTregs in their DKO mice, since pTregs has been shown to be influenced by intestinal bacteria.

(3) Ig levels and auto-antibody should be examined, since GC-B cells were increased in DKO mice.

(4) Authors showed high levels of Th17 and Tfh related gene expression on DKO Tregs. Is this due to reduced Foxp3 expression? Can these phenotypes be diminished by Foxp3 overexpression? On the contrary, unstable Foxp3 expression may be due to high levels of Bcl6 or Rorc in DKO Tregs. This point can be addressed by knocking down of these genes using CRISPR or shRNA in DKO Tregs, then transfer into Rag-deficient mice. This is a bit difficult experiment since authors have to introduce genes into Tregs, but worthwhile to try to define the mechanism of Th17 and Tfh

skewing.

Reviewer #2 (Remarks to the Author):

Yue et al studied the effects of conditional dual deletion of Tet2 and Tet3 in murine Foxp3+ Treg cells. This is a modest and incremental advance on their 2016 JEM paper in which these enzymes were deleted using CD4cre, and the 2015 Yang et al report in Immunity involving dual deletion of Tet1 and Tet2, and now includes some evidence of the development of a pathogenic ex-Treg population.

1. The work begins with characterization of mice with conditional Treg deletion of Tet2 and Tet3 (Fig. 1). This section is not well done. What age was the mouse that produced the data in Suppl. Fig. 1, and can the authors provide any insight into why the mesenteric lymph node was so enlarged compared negligible expansion of peripheral LNs, and moderate splenomegaly? Was CCR7 or L-selectin downregulated? Mean survival appear to be >100 days and despite supposed "massive lymphocyte infiltration" into the lung, these don't actually look that bad. A full necropsy should be performed and histopathology data reported by an experienced pathologist. What causes death of the mice? Usually in scurfy mice, despite all the published images of skin and lungs and liver infiltrates, it is BM failure due to tri-lineage injury that leads the mice to succumb. Hematologic data should be provided here, and were autoantibodies generated, including those capable of promoting hemolytic anemia?

2. The effects of Tet2/3 deletion warrant careful consideration (Fig. 2). What age were these mice? Effects on the % of Tregs are shown for various sites but absolute numbers should also be provided. The subsequent effects on Treg phenotype are probably better left until analysis of gene expression has occurred and thereby used to confirm events at the protein level. A sit stands, the markers chosen are but a small number of many proteins that could have been analyzed.

3. Next, we are shown data from an adoptive transfer model. This section on page 5 is entitled "Decreased long-term suppressive function of Tet2/3-deficient Treg cells" and yet later in the paragraph we are told that this is a "short-term in vivo suppression assay"; according to the authors, which is it? Could the data in Suppl. Fig. 2 reflect altered Treg trafficking after transfer as compared to WT Tregs? What happens in mucosal LNs? What is the function of the DKO Tregs like when compared to WT Tregs in a standard in vitro Treg assay using CFSE-labeled proliferating conventional T cells? Are there differences if Tregs are isolated from mesenteric LNs vs. other sites? Is iTreg development especially impaired in these mice, with a particular deficit at the level of mesenteric LN T cells?

4. The data re TSDR demethylation is not well handled (Fig. 3), and as often seems to be required to be asked in this paper, how old were the mice? If the Tet enzymes promote conversion of 5mC to 5hmC, why do the authors use the readout of 5mC+5hmC? Why not provide data on methylation or demethylation of specific CpG sites at CNS1 and especially CNS2, as is usually done?

5. The RNA-seq data (Fig. 4), which needs to be deposited in a public database, is pretty messy and potentially quite misleading given the DKO vs. DKO* issue. Why did the authors not compare events in DKO mice with similar splenomegaly, and why did they not analyze WT vs. DKO mesenteric LN Tregs as a key point rather than, apparently, pooling Tregs from spleens and LNs?

6. How do the authors explain the differences between data in Suppl. Fig. 1b and Fig. 5? Are the females in Suppl. Fig. 1a homozygous for Foxp3cre but heterozygous in Fig. 5? The ex-Treg data (Figures 5-7) is both consistent with previous reports and yet inconclusive, since as the authors note: "These might be ex-Treg effector cells, host CD4+ T cells that became activated and

converted to effector cells, or both”.

Reviewer #3 (Remarks to the Author):

This is an excellent study addressing the role of active demethylation in Treg function, which extends prior work in this area by specifically deleting Tet2 and Tet3 in the Treg lineage using Foxp3-Cre. The topic is of high significance, and the authors' conclusions are supported by the data presented. I have a few comments and a concern that, if addressed, could lead to an improved manuscript.

The manuscript focuses on the role of Tet enzymes in methylation/hydroxy-methylation at the Foxp3 locus, which is clearly important, but promotes a somewhat myopic view that the role of Tet proteins in Treg is exclusively to regulate Foxp3 expression/stability. However, DNA methylation patterns must certainly be dysregulated genome-wide in these cells - to what extent does DNA methylation resemble that of conventional and/or effector T cells at other loci, or to what extent is DNA methylation further dysregulated beyond patterns in effector cells? Any additional analyses would lend further insight into the role of Tet proteins in Treg function. This also raises the possibility that dysregulated Tet function may be toxic - while Foxp3⁺ Treg are clearly present in Tet2/3-deficient animals, can the authors provide any data on the rate of loss of Treg in these animals to apoptosis vs. loss to ex-Treg? Assaying the frequency of apoptotic Foxp3⁺ cells in lymph nodes or spleen by TUNEL could potentially address this.

The dominant nature of the inflammatory phenotype in Treg-specific Tet2/3-deficient animals - i.e., the inability of wild-type Treg to extrinsically suppress inflammatory disease in animals that also contain Tet2/3-deficient Treg - is striking, and the authors interpret this as a result of the increased rate of Foxp3 loss by the ko Treg. This interpretation would necessitate that loss of Tet2/3 renders ex-Treg resistant to wild-type Treg-mediated suppression, but this is not specifically addressed. I suggest two approaches that could specifically address this point - 1) the authors could sort ex-Tregs from Tet2/3 wild-type tracer mice and from Tet2/3-deficient tracer mice, and assess the capacity of these two ex-Treg populations to mediate disease if co-transferred with wild-type Treg. This could be done in the scurfy model used in the current manuscript, or in an adoptive transfer colitis model. Alternatively or in addition, 2) the authors could ectopically force Foxp3 expression in wild-type vs. Tet2/3-deficient CD4 T cells in order to assess the Foxp3-independent effects of Tet2/3 loss on Treg function. This experiment has the power to assess the potential pathologic capacity of Foxp3⁺ Tet2/3-deficient Treg uncoupled from their increased capacity to lose Foxp3 expression.

Responses to reviewers' comments:

Reviewer #1

This paper by Yue et al. analyzed the role of Tet2/Tet3 in Tregs by generating Treg-specific Tet2/3-double deficient (DKO) mice. They found that DKO mice develop lymphoproliferative and inflammatory diseases. Both Foxp3⁺ and Foxp3⁻ cells showed strong skewing to Tfh and/or Th17 phenotypes. Treg cells from DKO mice were more prone to lose Foxp3 expression and lost suppression activity in vivo. Authors concluded that Tet2 and Tet3 are guardians of Treg cell stability and immune homeostasis in mice.

This paper extend their previous reports showing unstable Foxp3 expression in Tet2/Tet3 DKO Tregs (Yue et al. J. Exp. Med. 2016). In addition, they discovered a strong phenotype of Th17/Tfh skewing and fatal lymphoproliferative disease in their mice. They discovered altered expression Th17/Tfh genes, cell cycle-related genes and DNA damage repair genes in Tregs in their DKO mice. These phenotypes are very interesting, even though the mechanism is still open question. I have several suggestions which may improve the value of the paper.

Response: We are happy that the reviewer finds our results of interest.

(1) Previously authors showed that TCR-dependent iNKT cell expansion in T cell-specific (CD4Cre)-DKO mice. Likewise Foxp3⁺Tregs were expanded in their Foxp3Cre-DKO mice. In Supplementary Fig.4, authors showed RNA-seq data of CDR3 suggesting a clonal expansion in both Treg cells and CD4⁺Foxp3⁻ cells. It is not very clear whether expansion of Foxp3⁻ cells is due to "bystander expansion" or "expansion of exFoxp3 cells". Therefore, it may be better if authors can show the actual sequences of CDR3 from DKO Foxp3⁺ and Foxp3⁻ cells.

Response: We have now included the actual CDR3 sequences as **NEW** Table S6 and S7.

In addition, it is informative to check whether expanded Foxp3⁻ cells lacked Tet2/Tet3 genes or not. If expansion of Foxp3⁻ cells is a bystander effect, how authors explain this? High levels of IL-2?

Response: We have observed using qPCR that Tet2 and Tet3 mRNA expression levels are considerably reduced in CD4⁺YFP⁻ T cells isolated from Tet2/3 DKO compared to WT mice (**NEW** Supplementary Figure 12a). Consistent with this, our RNA-seq data show that the number of normalized reads located in the deleted exons for both Tet2 and Tet3 was also decreased in CD4⁺YFP⁻ cells isolated from Tet2/3^{fl/fl}Foxp3^{Cre} compared to WT mice (**NEW** Supplementary Figure 12b-c). Thus both qPCR and RNA-seq data show that Tet2/3 DKO mice display expansion of ex-Treg cells which previously expressed Foxp3 but had lost Foxp3 expression at the time of analysis. In addition, **NEW** Figure 7 shows that when we transferred total CD4⁺ T cells from Tet2/3 DKO mice into immunocompetent recipient mice, the host cells also became activated (see Figure 7c, panels for "Host Cells"), suggesting that bystander CD4⁺ T cell expansion also occurred in Tet2/3 DKO mice.

Based on these data, we conclude that the expansion is due both to "expansion of ex-Treg cells" and to "bystander CD4⁺ T cell expansion". And the bystander expansion may be mediated through cytokine secretion, for example the increased expression of IL17 cytokines.

(2) Similarly, Foxp3⁻cells tend to become Th17/Tfh-like cells due to high levels of IL-6 or IL-21? This can be addressed in vitro culture of Tregs with these cytokines and measurement of Th17/Tfh phenotypes.

Response: As suggested by the reviewer, we sorted CD4⁺YFP⁺ Treg cells, cultured them in vitro with TCR stimulation and IL2 in the presence of different concentrations of the cytokines IL-6 and IL-21 (10 ng/ml and 100 ng/ml) for three days, then measured Th17 (IL17A, IL17F and Ror γ t expression) and Tfh (PD1 and CXCR5 expression) phenotypes without further restimulation. We observed that in the presence of IL6 and IL21, expression levels of the Th17-related molecules IL17F and Ror γ t were increased in both WT and DKO Treg cells, with somewhat more pronounced effects on DKO Treg cells (**Figure 1a for the reviewers**), but expression levels of CXCR5 and PD1 were unchanged (**Figure 1b for reviewers**).

(3) Previous paper using CD4Cre-DKO indicates that TET-mediated CNS2 demethylation is necessary for maintenance of iTregs in vivo. Stable iTregs after transfer showed CNS2 demethylation in a TET-dependent manner.

Response: In our previous study, we showed that culture of differentiating iTregs with the TET activator Vitamin C resulted in CNS2 demethylation, and that these TGF β and Vitamin C-induced iTregs were more stable after adoptive transfer than iTregs induced with TGF β alone. The effect of Vitamin C was TET-dependent, since addition of Vitamin C did not cause CNS2 demethylation in iTregs generated from naïve CD4⁺ T cells of *Tet2/3* DKO mice. Thus the reviewer's statement that "Stable iTregs after transfer showed CNS2 demethylation in a TET-dependent manner" is not entirely correct.

Authors should check stability and CNS2 methylation of Foxp3 in iTregs from Foxp3CreDKO mice after transfer. This experiment may define the essential and/or redundant role of TETs in Foxp3 stability in mature Tregs.

Response: As mentioned above, we already showed in our previous study using *Tet2/3^{fl/fl} CD4^{Cre}* mice that CNS2 is not efficiently demethylated and that Foxp3 expression is not stable in iTregs generated without Vitamin C. That is, iTregs generated in culture with TGF β alone do not show CNS2 demethylation; if Vitamin C is included as a TET activator, robust demethylation is observed. Moreover, the effect of Vitamin C is TET-dependent, since Vitamin C did not cause CNS2 demethylation in iTregs lacking *Tet2* and *Tet3*. In contrast, *Tet2* and *Tet3* are deleted only as Foxp3 becomes expressed during iTreg differentiation of *Tet2/3^{fl/fl} Foxp3^{Cre}* cells. Overall, therefore, the experiments suggested by the reviewer may not improve our understanding of the relation between TET activity, CNS2 demethylation and Foxp3 stability beyond what we have already observed using *Tet2/3^{fl/fl} CD4^{Cre}* mice. We note that in addition to our study (Yue et al., 2016, PMID: 26903244), several other publications have suggested a correlation between TSDR demethylation and Foxp3 stability (Floess et al., 2007 PMID: 17298177; Toker et al., 2013, PMID: 23420886).

(4) Figs.3/4: Unstable Foxp3 expression is probably due to reduced demethylation of CNS1 and CNS2. However, methylation status of other TSDRs or Treg related genes seems not to be examined. Are there any correlations between Foxp3 instability/signature genes expression and their (TSDR) methylation status?

Response: We have now performed amplicon sequencing using the Miseq platform for the methylation status of other TSDRs including *Il2ra* intron 1a, *Tnfrsf18* exon 5, *Ikzf4* intron 1b and *Ctla4* exon 2. We used the same genomic DNA as shown previously in Figure 3, the mice used were 8-10 weeks old, and CD4⁺CD25^{hi}YFP⁺ cells were sorted for the analysis. We used the mice at younger age prior to the signs of cell expansion so that we can avoid any secondary effects due to the inflammation. The results are shown in the **NEW** Figure 3 c-f.

Minor comments

(1) DNA damages stress has been shown to be associated with malignant proliferation of Tets-deficient cells. How about DNA damages (γ -H2AX) in Foxp3+ and Foxp3- cells?

Response: We have now examined the expression level of γ -H2AX by flow cytometry, and the results show that both CD4⁺Foxp3⁺ Treg cells and CD4⁺Foxp3⁻ T cells from *Tet2/3* DKO mice have moderately increased levels of γ -H2AX compared to cells from WT mice (**NEW** Supplementary Figure 8).

(2) Authors mentioned the work by Bana Jabri (Nature, in press), which shows the importance of commensal bacteria for the phenotypes of Tet2-deficient mice. It will be informative if authors examine pTregs in their DKO mice, since pTregs has been shown to be influenced by intestinal bacteria.

Response: Both *Nrp1* and *Helios* have been reported as markers of thymic-derived Treg cells, therefore can be used to distinguish thymic-derived Treg cells from peripherally-induced Treg cells. However, the reliability of these markers is still controversial. We have now analyzed *Helios* and *Nrp1* expression levels in mesenteric lymph nodes by flow cytometry as shown in the **NEW** Supplementary Figure 3c. The expression of *Nrp1* was downregulated in Treg cells from mesenteric lymph nodes of *Tet2/3^{fl/fl} Foxp3^{Cre}* mice compared to WT mice, while the expression of *Helios* was increased.

(3) Ig levels and auto-antibody should be examined, since GC-B cells were increased in DKO mice.

Response: We now have analyzed immunoglobulin levels by Legendplex immunoglobulin panel and anti-dsDNA antibody production by ELISA, and these data are included as the **NEW** Supplementary Figure 2.

(4) Authors showed high levels of Th17 and Tfh related gene expression on DKO Tregs. Is this due to reduced Foxp3 expression? Can these phenotypes be diminished by Foxp3 overexpression? On the contrary, unstable Foxp3 expression may be due to high levels of Bcl6 or Rorc in DKO Tregs. This point can be addressed by

knocking down of these genes using CRISPR or shRNA in DKO Tregs, then transfer into Rag-deficient mice. This is a bit difficult experiment since authors have to introduce genes into Tregs, but worthwhile to try to define the mechanism of Th17 and Tfh skewing.

Response: We thank the reviewer for this suggestion. Treg cells from DKO mice showed increased frequencies of Foxp3-expressing cells and increased MFIs of Foxp3 expression as shown by flow cytometry (Figure 2a-2b), therefore the skewed Th17 and Tfh phenotype are not likely to be due to reduced Foxp3 expression. Technically, retroviral transduction of Treg cells and long-term culture of Treg cells in vitro are still quite challenging, we are currently working on conquering these technical barriers.

Reviewer #2

Yue et al studied the effects of conditional dual deletion of Tet2 and Tet3 in murine Foxp3+ Treg cells. This is a modest and incremental advance on their 2016 JEM paper in which these enzymes were deleted using CD4-Cre, and the 2015 Yang et al report in Immunity involving dual deletion of Tet1 and Tet2, and now includes some evidence of the development of a pathogenic ex-Treg population.

1. The work begins with characterization of mice with conditional Treg deletion of Tet2 and Tet3 (Fig. 1). This section is not well done. What age was the mouse that produced the data in Suppl. Fig. 1, and can the authors provide any insight into why the mesenteric lymph node was so enlarged compared negligible expansion of peripheral LNs, and moderate splenomegaly?

Response: We have now added information on the age of the mice used for each experiment into each figure legend. The mice used in Supplementary Figure 1 were 13-16 weeks old. The images shown in Supplementary Figure 1c are representative images from 14-week old WT and *Tet2/3^{fl/fl}Foxp3^{Cre}* mice. As shown in Figure 1b, the cellularity of spleen and mesenteric lymph nodes was significantly increased in *Tet2/3^{fl/fl}Foxp3^{Cre}* mice compared to WT mice, the cellularity of peripheral lymph nodes was also increased but did not reach statistical significance. As now discussed in detail in the revised Discussion, we believe that the most likely reason for the large increase in the size and cellularity of mesenteric compared to other peripheral lymph nodes is that TET deficiency promotes increased antigen- and/or cytokine-dependent expansion.

Was CCR7 or L-selectin downregulated?

Response: Flow cytometry analyses showed that the expression level of both CCR7 and CD62L (L-selectin) was significantly downregulated in *Tet2/3* DKO Treg cells (**NEW** Figure 2e, **NEW** Supplementary Figure 3d).

Mean survival appear to be >100 days and despite supposed "massive lymphocyte infiltration" into the lung, these don't actually look that bad. A full necropsy should be performed and histopathology data reported by an experienced pathologist.

Response: We agree and have removed the words "massive lymphocyte infiltration" from the text, replacing it with "leukocyte infiltration". Prof. Nissi Varki, a pathologist at UCSD, has assessed the histopathology of WT and *Tet2/3^{fl/fl}Foxp3^{Cre}* mice. She helped us perform immunohistochemistry with anti-CD3, anti-B220 and anti-F4/80 antibodies for samples isolated from spleen and mesenteric lymph nodes from *Tet2/3* DKO mice and age-matched WT controls. The results suggested that the splenomegaly and enlarged mesenteric lymph nodes were not the usual T-cell, B-cell tumors or histiocytic, instead it is highly likely to be a myeloid neoplasm. This was also supported by our flow cytometry data, showing that in *Tet2/3* DKO mice, the percentage of Gr1⁺CD11b⁺ cells increased significantly (**NEW** Supplementary Figure 4). We also reword our statement, stating now in the main text "Histological analysis revealed disrupted splenic architecture in *Tet2/3^{fl/fl}Foxp3^{Cre}* mice with expansion of the white pulp areas, accompanied by leukocyte infiltration into the lung." The leukocyte infiltration into the liver in *Tet2/3* DKO mice was very similar compared to that in the WT mice.

What causes death of the mice? Usually in scurfy mice, despite all the published images of skin and lungs and liver infiltrates, it is BM failure due to tri-lineage injury that leads the mice to succumb. Hematologic data should be provided here...

Response: We do not know precisely what causes the death of the mice. To address this point, we have provided hematologic data for *Tet2/3^{fl/fl}Foxp3^{Cre}* mice and age-matched WT control mice using Hemavet 950FS (**NEW** Table S1). We did not observe any signs of anemia.

...were autoantibodies generated, including those capable of promoting hemolytic anemia?

We examined the level of anti-dsDNA autoantibody by ELISA and serum immunoglobulin levels by Legendplex immunoglobulin panel (**NEW** Supplementary Figure 2) and the results showed that *Tet2/3^{fl/fl}Foxp3^{Cre}* mice had significantly higher titers of anti-dsDNA antibodies and IgG2b isotype in the serum compared to WT mice.

2. The effects of *Tet2/3* deletion warrant careful consideration (Fig. 2). What age were these mice? Effects on the % of Tregs are shown for various sites but absolute numbers should also be provided.

Response: We appreciate the reviewer's comment on this point. We have now added information on the age of the mice used for each experiment into each figure legend. The mice used in Figure 2 were 13-16 weeks old. We also added graphs showing the absolute numbers of Treg cells in the **NEW** Figure 2b (right panel).

The subsequent effects on Treg phenotype are probably better left until analysis of gene expression has occurred and thereby used to confirm events at the protein level. As it stands, the markers chosen are but a small number of many proteins that could have been analyzed.

Response: We hope that the reviewer agrees that the markers we chose in the **NEW** Figure 2c-2d are well-known Treg cell associated markers. We did not perform RNA-seq for all samples because as we show in Figure 4, the data are influenced by the severity of the disease phenotype in the mice (see our response to point 5 below), and because gene expression changes at the transcriptional level do not always correlate with changes at the protein level.

3. Next, we are shown data from an adoptive transfer model. This section on page 5 is entitled "Decreased long-term suppressive function of *Tet2/3*-deficient Treg cells" and yet later in the paragraph we are told that this is a "short-term *in vivo* suppression assay"; according to the authors, which is it? Could the data in Suppl. Fig. 2 reflect altered Treg trafficking after transfer as compared to WT Tregs? What happens in mucosal LNs?

Response: In supplementary Figure 5a-c, we showed the adoptive transfer model to assess the capability of *Tet2/3* DKO Treg cells to correct the scurfy phenotype *in vivo*. We analyzed the recipient mice 4-5 weeks after adoptive transfer, therefore we call this "short-term" and in this setting, Treg cells isolated from *Tet2/3^{fl/fl}Foxp3^{Cre}* mice suppressed the expansion of scurfy CD4⁺ T cells to almost the same extent as Treg cells isolated from WT *Foxp3^{Cre}* mice. In Supplementary Figure 5d-e, we showed another adoptive transfer experiment, where we observed the recipient mice for longer time to examine the survival of the recipient mice and in this setting, co-transfer of scurfy bone marrow together with *Tet2/3* DKO bone marrow cells failed to protect the recipient mice. Therefore, we concluded that *Tet2/3*-deficient Treg cells from *Tet2/3^{fl/fl}Foxp3^{Cre}* mice have normal *in vivo* suppressive function in the short-term that is not sustained in the long-term. We have now also included the data from mesenteric LNs into the **NEW** supplementary Figure 5c, bottom panel.

*What is the function of the DKO Tregs like when compared to WT Tregs in a standard *in vitro* Treg assay using CFSE-labeled proliferating conventional T cells? Are there differences if Tregs are isolated from mesenteric LNs vs. other sites? Is iTreg development especially impaired in these mice, with a particular deficit at the level of mesenteric LN T cells?*

We have now shown in a standard *in vitro* suppression assay that *Tet2/3* DKO Treg cells are less suppressive compared to WT Treg cells (see Figure 2 for reviewers). We have also performed RNA-seq for Treg cells isolated from mesenteric lymph nodes from WT and *Tet2/3^{fl/fl}Foxp3^{Cre}* mice (see responses to reviewer 2's comment 5 below). We did not analyze iTreg development in these mice since *Tet2* and *Tet3* are specifically deleted only in Treg cells (i.e. only after Foxp3 has been induced), not in other T cell compartments.

4. The data re TSDR demethylation is not well handled (Fig. 3), and as often seems to be required to be asked in this paper, how old were the mice?

Response: The mice used for the analysis of TSDR methylation status were 8-10 weeks old, we have now included this information in the figure legend of Figure 3.

If the Tet enzymes promote conversion of 5mC to 5hmC, why do the authors use the readout of 5mC+5hmC? Why not provide data on methylation or demethylation of specific CpG sites at CNS1 and especially CNS2, as is usually done?

The bisulfite sequencing we have performed is exactly the same analysis as is usually reported by other groups. However, bisulfite sequencing does not distinguish 5hmC from 5mC (see Huang et al, 2010, PMID: 20126651) and hence the common description of bisulfite sequencing data as showing methylation (i.e. 5mC)

status is inaccurate. Rather, bisulfite treatment yields the percentage of non-converted Cs – corresponding to total 5mC+5hmC – at each analysed CpG. Thus our labelling of the data from bisulfite sequencing as (5mC+5hmC) is precise and accurate. It is true that TET-deficient cells will have less or no 5hmC compared to WT cells, and therefore the signal from bisulfite sequencing in TET-deficient cells will be mostly 5mC.

5. The RNA-seq data (Fig. 4), which needs to be deposited in a public database, is pretty messy and potentially quite misleading given the DKO vs. DKO* issue. Why did the authors not compare events in DKO mice with similar splenomegaly?

Response: The RNA-seq data were indeed deposited into the GEO series database (GSE113694); this information was included in the *Methods* section. For RNA-seq analysis, we intentionally did not use mice with similar splenomegaly because we wished to emphasize that *Tet2/3* DKO mice of the same age often show very different disease severities, and because splenomegaly and other parameters of disease severity do not reflect the immediate effect of *Tet2/3* deletion.

Why did they not analyze WT vs. DKO mesenteric LN Tregs as a key point rather than, apparently, pooling Tregs from spleens and LNs?

Response: We have now performed RNA-seq analysis for CD4⁺YFP⁺ Tregs and CD4⁺YFP⁻ T cells (a mixture of ex-Tregs and bystander CD4⁺ T cells) isolated from mesenteric lymph nodes (mLN). The data are now included in the **NEW** supplementary Figure 7. Based on a cut-off of FDR (false discovery rate) ≤ 0.05 and a fold change ≥ 1.5 , 2073 genes were differentially expressed, with 1449 genes being upregulated and 624 genes being downregulated in *Tet2/3* DKO Treg cells compared to WT Treg cells from mLNs (**NEW** Supplementary Fig. 7a). A handful of them overlapped with the differentially expressed genes from *Tet2/3* DKO vs WT Treg cells isolated from pooled spleen and pLN (**NEW** Supplementary Fig. 7b). Ingenuity Pathway Analysis (IPA) analysis of the differentially expressed genes indicated that similar canonical pathways were affected, including genes involved in cell cycle, DNA damage, immune cell function and cancer development (**NEW** Supplementary Fig. 7c and **NEW** Table S5).

6. How do the authors explain the differences between data in Suppl. Fig. 1b and Fig. 5? Are the females in Suppl. Fig. 1a homozygous for *Foxp3cre* but heterozygous in Fig. 5? The ex-Treg data (Figures 5-7) is both consistent with previous reports and yet inconclusive, since as the authors note: “These might be ex-Treg effector cells, host CD4⁺ T cells that became activated and converted to effector cells, or both”.

Response: The reviewer is right; Supplementary Figure 1b shows females homozygous for *Foxp3-Cre* and Figure 5a-b shows females heterozygous for *Foxp3-Cre*. Since *Foxp3* is X-chromosome encoded, *Tet2/3^{fl/fl}Foxp3^{Cre/Cre}* homozygous females will have YFP-Cre knocked in into both alleles and will delete *Tet2* and *Tet3* in all Treg cells, whereas *Tet2/3^{fl/fl}Foxp3^{WT/Cre}* heterozygous females will have one allele as the YFP-Cre knock-in, but the other allele will be wildtype and so only half the Treg cells will have *Tet2* and *Tet3* deleted.

Reviewer #3

*This is an excellent study addressing the role of active demethylation in Treg function, which extends prior work in this area by specifically deleting *Tet2* and *Tet3* in the Treg lineage using *Foxp3-Cre*. The topic is of high significance, and the authors' conclusions are supported by the data presented.*

Response: We are happy that the reviewer finds our results of interest.

I have a few comments and a concern that, if addressed, could lead to an improved manuscript.

*The manuscript focuses on the role of Tet enzymes in methylation/hydroxy-methylation at the *Foxp3* locus, which is clearly important, but promotes a somewhat myopic view that the role of Tet proteins in Treg is exclusively to regulate *Foxp3* expression/stability. However, DNA methylation patterns must certainly be dysregulated genome-wide in these cells - to what extent does DNA methylation resemble that of conventional and/or effector T cells at other loci, or to what extent is DNA methylation further dysregulated beyond patterns in effector cells? Any additional analyses would lend further insight into the role of Tet proteins in Treg function.*

Response: We thank the reviewer for these suggestions. We are currently performing a genome-wide analysis of the relation among DNA methylation, 5hmC distribution, chromatin accessibility and gene expression patterns in *Tet2/3*-deficient Treg and iTreg cells. Since changes in DNA methylation patterns from *Foxp3-Cre*

DKO mice are intermediate between Treg cells from WT mice and CD4-Cre DKO mice, we decided to use CD4-Cre DKO mice to study these global changes in DNA methylation and other genome-wide features due to *Tet2/3* deletion. To examine changes in DNA methylation patterns in *Tet2/3^{fl/fl}Foxp3^{Cre}* mice beyond *Foxp3* *CNS1* and *CNS2*, we have now included amplicon sequencing analysis of other previously reported TSDRs, including *Il2ra* intron 1a, *Tnfrsf18* exon 5, *Ikzf4* intron 1b and *Ctla4* exon 2 (**NEW** Figure 3c-f).

This also raises the possibility that dysregulated Tet function may be toxic - while Foxp3+ Treg are clearly present in Tet2/3-deficient animals, can the authors provide any data on the rate of loss of Treg in these animals to apoptosis vs. loss to ex-Treg? Assaying the frequency of apoptotic Foxp3+ cells in lymph nodes or spleen by TUNEL could potentially address this.

To assess Treg cell apoptosis, we now performed flow cytometry staining for active caspase-3 (**NEW** Supplementary Figure 9) and the percentage of active-caspase3 positive cells was very comparable for Treg cells from *Tet2/3^{fl/fl}Foxp3^{Cre}* mice and WT mice.

The dominant nature of the inflammatory phenotype in Treg-specific Tet2/3-deficient animals - i.e., the inability of wild-type Treg to extrinsically suppress inflammatory disease in animals that also contain Tet2/3-deficient Treg - is striking, and the authors interpret this as a result of the increased rate of Foxp3 loss by the ko Treg. This interpretation would necessitate that loss of Tet2/3 renders ex-Treg resistant to wild-type Treg-mediated suppression, but this is not specifically addressed. I suggest two approaches that could specifically address this point - 1) the authors could sort ex-Tregs from Tet2/3 wild-type tracer mice and from Tet2/3-deficient tracer mice, and assess the capacity of these two ex-Treg populations to mediate disease if co-transferred with wild-type Treg. This could be done in the scurfy model used in the current manuscript, or in an adoptive transfer colitis model.

Response: The tracer mice we have are *Tet2^{fl/fl}Tet3^{fl/fl}Foxp3^{YFP-Cre}Rosa26-YFP⁺* mice. Since YFP expression from the *Rosa26* locus is much brighter than YFP expression from the YFP-Cre fusion protein introduced into the 3' UTR of the *Foxp3* gene, cells expressing YFP from the *Rosa26* locus can be easily distinguished from cells expressing the YFP-Cre fusion protein encoded in the *Foxp3* gene. We can use these mice to examine the percentage of ex-Tregs by cell sorting of YFP^{high} cells and intracellular Foxp3 staining, but cannot sort the live ex-Treg cells. To address this question rigorously, we are now in the process of breeding *Tet2^{fl/fl}Tet3^{fl/fl}Foxp3^{YFP-Cre}* mice with *Rosa26-Tdtomato* reporter mice, but mice of the correct genotype will not be available for several more months.

As an alternative approach, we sorted YFP^{high} cells from heterozygous *Tet2^{+fl}Tet3^{+fl}Foxp3^{YFP-Cre}Rosa26-YFP⁺* mice and DKO *Tet2^{fl/fl}Tet3^{fl/fl}Foxp3^{YFP-Cre}Rosa26-YFP⁺* mice, which contained ex-Treg cells. We then co-transferred these Treg cells with congenic CD45.1⁺ naïve T cells into Rag-deficient animals and analyzed the recipient mice 7-8 weeks after adoptive transfer. The results show that transfer of YFP^{high} cells from heterozygous mice led to increased spleen cellularity in recipient mice (Figure 3c-3d for reviewers) and three recipients all lost Foxp3 expression in transferred heterozygous YFP^{high} cells (from 68.3% to 6.4%, 0.1% and 26% respectively). YFP^{high} cells from DKO mice also led to increased cellularity in recipient mice except for the DKO #5 but to a lesser extent compared to that for heterozygous mice. Three recipients lost Foxp3 expression in transferred DKO YFP^{high} cells (from 12.3% to 9.2%, 3.1% and 1% respectively), while three other recipients, in contrast, gained Foxp3 expression (from 12.3% to 45.8%, 18% and 64.5% respectively).

Alternatively or in addition, 2) the authors could ectopically force Foxp3 expression in wild-type vs. Tet2/3-deficient CD4 T cells in order to assess the Foxp3-independent effects of Tet2/3 loss on Treg function. This experiment has the power to assess the potential pathologic capacity of Foxp3+ Tet2/3-deficient Treg uncoupled from their increased capacity to lose Foxp3 expression.

Response: The suggestion of ectopic force expression of Foxp3 in *Tet2/3* DKO Treg cells is excellent. However, technically, retroviral transduction of Treg cells and long-term culture of Treg cells in vitro are still quite challenging. We are currently working on conquering these technical barriers.

Figure 1 for Reviewers

Figure 1 for reviewers: In vitro culture of *Tet2/3* DKO Treg cells in the presence of IL6 and IL21.

CD4⁺YFP⁺ Treg cells were sorted from WT and *Tet2/3^{fl/fl}Foxp3^{Cre}* DKO mice (13-16 weeks old) and cultured in vitro with TCR stimulation and IL2 in the presence of different concentrations of cytokine rIL-6 and rIL-21 (10ng/ml and 100ng/ml) for three days and then measured (a) Th17 phenotype (IL17A, IL17F and Ror γ t expression) and (b) Tfh phenotype (PD1 and CXCR5 expression) without further restimulation. Data are from two independent experiments.

Figure 2 for Reviewers

Figure 2 for reviewers: In vitro suppression assay for WT and *Tet2/3* DKO Treg cells.

CD4⁺YFP⁺ Treg cells were sorted from WT and *Tet2/3^{fl/fl}Foxp3^{Cre}* DKO mice (11 weeks old) were cultured with VPD (Violet proliferation dye, BD Biosciences) labeled CD4⁺CD25⁻CD62L^{hi}CD44^{lo} naïve T cells from CD45.1⁺ congenic mice at the indicated suppressor to responder ratio and in the presence of irradiated T cell-depleted splenocytes from CD45.2⁺ B6 mice and anti-CD3 (300ng/ml) for three days. Protocol was adopted from Li et al., 2004 (PMID: 25126782). **a.** Representative flow cytometry plots for WT suppressor (upper panel) and *Tet2/3* DKO suppressor (lower panel), the percentage of undivided responder cells were labeled on each plot. **b.** The average percentage of undivided responder cells from two independent experiments.

Figure 3 for Reviewers

Figure 3 for reviewers: The effects of ex-Treg cells in a cell-transfer model.

1×10^5 CD45.2⁺YFP^{high} cells were sorted from mesenteric lymph nodes (mLNs) of heterozygous *Tet2/3^{+/fl}Foxp3^{Cre}Rosa26-YFP^{LSL}* and DKO *Tet2/3^{fl/fl}Foxp3^{Cre}Rosa26-YFP^{LSL}* mice (16 weeks old) and co-transferred together with 4×10^5 CD45.1⁺ congenic naïve T cells into RAG-deficient animals. The recipients were analyzed 7-8 weeks after adoptive transfer. **a.** The ratio of CD45.1⁺ naïve T cells to CD45.2⁺ Het or DKO YFP^{high} cells before cell transfer. **b.** The percentage of ex-Treg cells in the sorted Het or DKO YFP^{high} cells. **c.** The photos of the spleens from the recipient mice (3 recipients of Het YFP^{high} cells and 6 recipients for DKO YFP^{high} cells). **d.** The cellularity of spleens from the recipient mice. **e.** The flow cytometry analysis of 3 recipients of Het YFP^{high} cells for the percentage of CD45.1⁺ and CD45.2⁺ cells within the CD4⁺ T cells; the percentage of Foxp3⁺ cells within CD4⁺CD45.2⁺ cells and the percentage of Foxp3⁺ cells within CD4⁺CD45.1⁺ cells. **f.** The flow cytometry analysis of 6 recipients for DKO YFP^{high} cells for the percentage of CD45.1⁺ and CD45.2⁺ cells within the CD4⁺ T cells; the percentage of Foxp3⁺ cells within CD4⁺CD45.2⁺ cells and the percentage of Foxp3⁺ cells within CD4⁺CD45.1⁺ cells.

Reviewers' comments:

Reviewer #1 (Remarks to the Author):

Authors responded properly to my comennts. I think they also repsoned well to other reviewer's comments. I have no further comments.

Reviewer #2 (Remarks to the Author):

The revised paper, the third run at the same topic by the authors, is improved but still only incremental in scope. The effects of Tet2 and Tet3 deletion in Foxp3+ Tregs aren't nearly as severe as many previously reported mice (50% survival herein of ~100 days). The impact in terms of histopathology is marginal and the authors have now provided the point that much of the lymphadenopathy is "highly likely to be due to a myeloid neoplasm"!!!

I.e. We don't really know what is going on in these mice.

This huge flaw, plus the marked but unexplained differences between mesenteric and other lymphoid tissues, as well as the failure to really nail down what dysregulation underlies the overall phenotype, mean to me that the paper, while data-rich, is impact-poor and better suited to a specialist immunologic journal.

Reviewer #3 (Remarks to the Author):

The authors have addressed my concerns to a reasonable degree, and together with additional data provided in response to the other reviewers, the manuscript has been significantly improved

Responses to reviewers' comments (for the 2nd revision):

Reviewer #1

Authors responded properly to my comments. I think they also responded well to other reviewer's comments. I have no further comments.

Response: We are happy that the reviewer is satisfied with our revised manuscript, which indeed improved thanks to all the reviewers' comments.

Reviewer #2

The revised paper, the third run at the same topic by the authors, is improved but still only incremental in scope.

Response: Our lab has previously characterized CD4-Cre mediated Tet-deficient mice in which we saw both iNKT cell expansion and Treg dysfunction (Tsagaratou A et al., PMID 27869820; Yue et al., PMID 26903244). Although Treg cell function was affected in these mice, the phenotype was confounded by 1) simultaneous dysregulation of iNKT cells; 2) CD4-Cre expression prior to the development of thymic Treg cells. Since Treg cells are essential for immune homeostasis, it is imperative to address the function of Tet proteins specifically in regulatory T cells using the Treg-specific Cre recombinase as in this study.

The effects of Tet2 and Tet3 deletion in Foxp3+ Tregs aren't nearly as severe as many previously reported mice (50% survival herein of ~100 days).

Response: We are not entirely sure what the reviewer means. If the reviewer means that the effects of *Tet2* and *Tet3* deletion using Treg-specific Cre were not as severe as the effects of *Tet2* and *Tet3* deletion using CD4-Cre, which we showed in our previous publications, this is due to the fact that CD4-Cre is mediating *Tet2/3* deletion in all the cell types which have experienced CD4 expression, including CD4⁺ T cells, CD8⁺ T cells, CD4⁺Foxp3⁺ Treg cells and iNKT cells. Instead, Foxp3-Cre is mediating *Tet2/3* deletion specifically in Treg cells. In addition, this also reflects the chronological order of CD4-Cre versus Foxp3-Cre expression: CD4-Cre mediates *Tet2* and *Tet3* deletion at an earlier developmental stage compared to Foxp3-Cre.

But if the reviewer means that the effects of *Tet2* and *Tet3* deletion with Foxp3-Cre are not as severe as many reported mice with deletions of different genes, we think this is not really comparable. The main focus of our manuscript is to understand the role of Tet proteins in Treg cells using the Foxp3-Cre mediated *Tet2/3* DKO mice as an animal model, instead of pathologically understanding the cause of the death in the mice.

The impact in terms of histopathology is marginal and the authors have now provided the point that much of the lymphadenopathy is "highly likely to be due to a myeloid neoplasm"!!!

Response: We apologize for the improper wording of "myeloid neoplasm" in the first rebuttal letter; we intended to describe myeloid expansion. In the NEW Supplementary Figure 4 for the first revision, we showed that CD11b⁺Gr1⁺ myeloid cells expanded in Foxp3-Cre mediated *Tet2/3* DKO mice, but this is a cell-extrinsic expansion instead of neoplasm. We now included a **NEW Figure for reviewer** to support this conclusion. In mixed bone marrow chimeras reconstituted with CD45.1⁺ WT bone marrow cells and CD45.2⁺ *Tet2/3* DKO bone marrow cells at 1:1 ratio, both CD45.1⁺ and CD45.2⁺ cells showed similar expansion of CD11b⁺Gr1⁺ myeloid cells, suggesting that this is a cell-extrinsic expansion. Therefore, the expansion of myeloid cells in the Foxp3-Cre *Tet2/3* DKO mice was caused by the presence of Tet-deficient Treg or ex-Treg cells.

i.e. We don't really know what is going on in these mice.

Response: Our manuscript showed that 1) *Tet2/3^{fl/fl}Foxp3^{Cre}* mice developed an inflammatory disease with splenomegaly, tissue infiltration and activated phenotype in multiple cell types; 2) At the transcriptional level, Treg cells from *Tet2/3^{fl/fl}Foxp3^{Cre}* mice showed altered expression of Treg signature genes, with upregulation of genes involved in cell cycle regulation, DNA damage repair and cancer; 3) The presence of WT Treg cells in *Tet2/3^{fl/fl}Foxp3^{WT/Cre}* heterozygous female mice and mixed bone marrow chimeras was not sufficient to suppress the inflammatory phenotype observed in *Tet2/3* DKO mice; 4) Fate-mapping experiments showed that Treg cells from *Tet2/3^{fl/fl}Foxp3^{Cre}* mice were more prone to lose Foxp3 expression; 5) Transfer of total CD4⁺ T cells from *Tet2/3^{fl/fl}Foxp3^{Cre}* DKO mice, which contained these ex-Treg cells, elicited inflammatory disease in immunocompetent mice. Therefore it does not seem fair for the reviewer to comment "we don't really know what is going on in these mice."

This huge flaw, plus the marked but unexplained differences between mesenteric and other lymphoid tissues,

Response: In the discussion section, we discussed the potential cause of the differences in the mesenteric lymph nodes and peripheral lymph nodes (other lymphoid tissue, such as spleen, is also significantly enlarged). "...myeloid expansion in *Tet2*-deficient mice has been attributed, at least in part, to the influence of cytokines such as IL-6 produced in response to gut microbiota, since it is eliminated or greatly attenuated if the mice are housed in a germ-free facility⁴¹. Even in 12-16 week-old *Tet2/3^{fl/fl}Foxp3^{Cre}* mice, the large increase in the cellularity of mesenteric lymph nodes compared to peripheral lymph nodes (Fig. 1b, Supplementary Fig. 1c) and the increased clonality of both Foxp3⁺ Tregs and CD4⁺Foxp3⁻ T cells (ex-Treg cells and/or activated CD4⁺ bystander T cells) (Supplementary Fig. 10) suggests a role for recognition of microbial or other antigens." See the second paragraph of the discussion section in the main text.

as well as the failure to really nail down what dysregulation underlies the overall phenotype, mean to me that the paper, while data-rich, is impact-poor and better suited to a specialist immunologic journal.

Response: This comment is better judged by the Editors. *Nature Communications* is a multi-disciplinary journal which publishes many types of papers including immunology articles. Given the importance of regulatory T cells in various biological disciplines including immunology and cancer, we believe our study will be well received by wide-range of readers and is suitable for *Nature Communications*.

Reviewer #3

The authors have addressed my concerns to a reasonable degree, and together with additional data provided in response to the other reviewers, the manuscript has been significantly improved.

Response: We are happy that the reviewer is satisfied with our revised manuscript, which indeed improved thanks to all the reviewers' comments.

Figure for Reviewer 2

Figure for the Reviewer. Cell-extrinsic expansion of CD11b⁺Gr1⁺ myeloid cells in *Tet2/3* DKO mixed bone marrow chimeras.

Flow cytometry staining of splenocytes from WT mixed bone marrow chimeras (*left panels*) and *Tet2/3* DKO mixed bone marrow chimeras (*right panels*) at 18-20 weeks after adoptive transfer for the cell surface markers CD11b and Gr1 in CD45.1⁺ WT cells and CD45.2⁺ WT or *Tet2/3* DKO cells (gated on CD4⁺CD8⁻B220⁻ cells).

REVIEWERS' COMMENTS:

Reviewer #2 (Remarks to the Author):

Yue et al find that mice whose Tregs lack Tet2 and Tet3 eventually develop evidence of autoimmunity. The phenotype of targeted mice is much weaker than that of many others when deleted within Tregs, and their overall findings are simply an incremental advance over what is already known from the literature. They do not undertake gene targeting in adult mice such that we don't know if these events are purely of developmental relevance and, given the lack of their ability to directly target the gene products in WT mice, they aren't of direct translational significance. Hence, I very much doubt that this paper will be of broad interest to the readership.

Responses to reviewers' comments (for the final revision):

Reviewer #2

Yue et al find that mice whose Tregs lack Tet2 and Tet3 eventually develop evidence of autoimmunity. The phenotype of targeted mice is much weaker than that of many others when deleted within Tregs, and their overall findings are simply an incremental advance over what is already known from the literature.

The reviewer has two main concerns: *(i) “The phenotype of targeted mice is much weaker than that of many others when deleted within Tregs” and (ii) “the findings we report are “an incremental advance over what is already known from the literature”.*

We respectfully disagree with the reviewer’s opinions: we do not agree that the phenotype is weak or that our findings are an incremental advance. In this manuscript, we describe a dramatic Treg-deficient phenotype that has not been previously described: TET-deficient Treg cells can cause a dominant pathological phenotype that is not rescued in the presence of wildtype Treg cells, as in heterozygous *Foxp3-Cre* TET-deficient female mice as well as in mixed bone marrow chimeras. This is rare in the Treg literature: even scurfy mice that completely lack *Foxp3* and have no Treg cells can be rescued by the introduction of wildtype Treg cells. We have traced this finding to the presence of effector ex-Treg cells that transfer the pathology, and show that the pathology can also be transferred to immunocompetent recipient mice merely by transferring CD4⁺ T cells which contain the ex-Treg cells.

They do not undertake gene targeting in adult mice such that we don’t know if these events are purely of developmental relevance and, given the lack of their ability to directly target the gene products in WT mice, they aren’t of direct translational significance. Hence, I very much doubt that this paper will be of broad interest to the readership.

First, we would like to point out that the *Foxp3-Cre* system bypasses the early stages of Treg cell development, since it only turned on once *Foxp3* is expressed. Second, we assume that by saying, “*They do not undertake gene targeting in adult mice*”, the reviewer means that we should delete *Tet2* and *Tet3* inducibly in adult mice. This is an interesting idea which would require us to transfer *Tet2 fl/fl Tet3 fl/fl Cre-ERT2* Treg cells with a *Rosa26-YFP^{LSL}* reporter into new recipient mice, and monitor the effect on an autoimmune disease such as EAE, colitis or type I diabetes. In fact, we previously used *Cre-ERT2* to delete *Tet2* and *Tet3* inducibly using tamoxifen in the hematopoietic system (PMID: 26607761). However, this comment was not brought up by the reviewer previously; instead, it is an entirely new comment, made after we fully addressed all the issues raised by this reviewer and other reviewers previously. We appreciate the reviewer’s opinion but the suggestion is out of the scope of our comprehensive study, which has been approved by the other reviewers.